# Mature and migratory dendritic cells promote immune infiltration and response to anti-PD-1 checkpoint blockade in metastatic melanoma

Jiekun Yang [1,2,3,4,12] ✉, Cassia Wang[1,12], Doris Fu[1,12], Li-Lun Ho[1,2], Kyriakitsa Galani[1,2], Lee Chen [1], Jose Gonzalez[3,4], Jolene Fu[3,4], Amy Y. Huang [2,5,6], Dennie T. Frederick[7], Liang He[1,2], Mukta Asnani[3,4], Rahul Tacke[1], Emily J. Robitschek[2,5,6], Sandeep K. Yadav [8], Wentao Deng[8], Kelly P. Burke [2,5,6], Tatyana Sharova[9], Ryan J. Sullivan [6,10], Sarah Weiss [11], Kunal Rai [8], David Liu [2,5,6], Genevieve M. Boland [2,9] ✉ & Manolis Kellis [1,2] ✉

Immune checkpoint inhibitors (ICIs) have revolutionized cancer therapy, yet most patients fail to achieve durable responses. To better understand the tumor microenvironment (TME), we analyze single-cell RNA-seq (~189 K cells) from 36 metastatic melanoma samples, defining 14 cell types, 55 subtypes, and 15 transcriptional hallmarks of malignant cells. Correlations between cell subtype proportions reveal six distinct clusters, with a mature dendritic cell subtype enriched in immunoregulatory molecules (mregDC) linked to naive T and B cells. Importantly, mregDC abundance predicts progression-free survival (PFS) with ICIs and other therapies, especially when combined with the TCF7 + /− CD8 T cell ratio. Analysis of an independent cohort (n = 318) validates mregDC as a predictive biomarker for anti-CTLA-4 plus anti-PD-1 therapies. Further characterization of mregDCs versus conventional dendritic cells (cDC1/cDC2) highlights their unique transcriptional, epigenetic (single-nucleus ATAC-seq data for cDCs from 14 matched samples), and interaction profiles, offering new insights for improving immunotherapy response and guiding future combination treatments.

Cancer treatment has progressed significantly, moving beyond targeting highly proliferative cells to identifying precise targets within cancer cells and their specific immune microenvironment. Combination therapies, whether administered simultaneously or sequentially, are frequently used for increased efficacy in advanced cancer stages. The emergence of immune checkpoint inhibitors (ICIs) approximately a decade ago and their use in advanced melanoma revolutionized cancer treatment[1]. Prior to ICIs, patients with stage IV melanoma

[1]Computer Science and Artificial Intelligence Laboratory, Massachusetts Institute of Technology, Cambridge, MA, USA. [2]Broad Institute of MIT and Harvard, Cambridge, MA, USA. [3]Department of Genetics, School of Arts and Sciences, Rutgers University-New Brunswick, Piscataway, NJ, USA. [4]Human Genetics Institute of New Jersey, Rutgers University-New Brunswick, Piscataway, NJ, USA. [5]Dana-Farber Cancer Institute, Boston, MA, USA. [6]Harvard Medical School, Boston, MA, USA. [7]Division of Medical Oncology, Department of Medicine, Mass General Brigham, Boston, MA, USA. [8]Department of Genomic Medicine and MDACC Epigenomics Therapy Initiative, University of Texas MD Anderson Cancer Center, Houston, TX, USA. [9]Division of Gastrointestinal and Oncologic Surgery, Department of Surgery, Mass General Brigham, Boston, MA, USA. [10]Division of Hematology and Oncology, Department of Medicine, Mass General Brigham, Boston, MA, USA. [11]Medical Oncology, Rutgers Cancer Institute, New Brunswick, NJ, USA. [12]These authors contributed equally: Jiekun Yang, Cassia Wang, and Doris Fu. ✉e-mail: jackie.yang@rutgers.edu; GMBOLAND@MGH.HARVARD.EDU; manoli@mit.edu

typically had a median lifespan of 6-9 months. However, with ICIs, survival rates extended up to 6 years with anti-PD-1/anti-CTLA-4 combination therapy, albeit with a higher incidence of severe adverse events[2]. Despite these advancements, only a subset of patients, ranging from 45 to 58%, demonstrated objective responses to individual or combined ICI treatments[3]. ICI treatments, used as a second-line treatment option, showed objective responses in only 15 to 30% of patients[4]. Resistance to targeted therapies and ICIs has posed challenges to patients' progression-free survival (PFS), underscoring the urgent need for biomarkers to predict survival benefits across treatments and guide clinical decision-making.

Over the past decade, research in metastatic melanoma has identified several response-associated biological features, such as tumor mutational burden (TMB) and neoantigen load[5], alongside CD8 expression at the invasive margin[6]. However, their predictive abilities remain limited. Advancements in single-cell technology have facilitated a more nuanced understanding of the tumor microenvironment (TME) at single-cell resolution and its relationship with clinical responses[7–11]. Due to limitations in cell collection and technology costs, these studies have primarily focused on tumor and immune cells, particularly T cells. In addition to the number and spatial distribution of T cells, which predict clinical response to ICIs, the presence of TCF7 protein in CD8 + T cells has been shown to be critical[9]. These stem-like TCF7 + CD8 + T cells maintain their stemness by expressing adult stem cell genes[12]. T cell factor 1 (TCF1, encoded by *TCF7*) has been reported to optimize tumor-specific CD8 + T cell priming in low-antigenic environments[13]. However, the role of myeloid cells, particularly dendritic cells, in influencing and determining T cell states such as TCF7 + CD8 + T cells in human tumors in vivo remains unclear.

Dendritic cells (DCs) play a crucial role in coordinating immune responses against tumors, serving as professional antigen-presenting cells (APCs) since their discovery in 1973[14]. Categorized into conventional DCs (cDCs) and plasmacytoid DCs (pDCs), cDCs further split into type 1 (cDC1) and type 2 (cDC2) based on lineage-determining TFs and functions[15–17]. While immature cDCs are essential for antigen capture and processing, their maturation is vital for immune surveillance involving antigen presentation and T-cell co-stimulation[18]. Despite their importance, DC signatures have been obscured in tumor measurements due to its low abundance[14], until recent single-cell sequencing efforts unveiled their heterogeneity, highlighting a regulatory signature termed mature DC enriched in immunoregulatory molecules (mregDCs)[19]. Although prior studies associated a higher density of mregDC-like DCs with improved survival in cutaneous melanoma[20,21], this state remains poorly characterized in terms of its epigenetic profile and interactions with T cells compared to other DC populations, especially in a clinical treatment setting.

To address these gaps, in this work, we conduct extensive single-cell profiling of metastatic melanoma samples, annotating diverse cell types and subtypes in the TME and correlating them with treatment response and patient survival. Analyzing 189,362 cells across 36 samples, we identify 14 cell types and 55 subtypes, shedding light on correlated subtypes and prognostic markers for immune checkpoint inhibitors (ICIs). Of particular interest are mregDCs, which emerge as survival predictors. We delve into their transcriptome signatures, epigenome landscape (single-nucleus ATAC-seq data for cDCs from 14 matched samples), differentiation trajectory, and cell-cell communication potentials, uncovering intrinsic and extrinsic molecules associated with mregDC abundance in the TME.

## Results
### Cell types in metastatic melanoma tumor microenvironment
We obtained metastatic melanoma samples from Mass General Brigham, focusing on annotated cases that had received various primary treatments, predominantly involving ICIs. After rigorous quality control, our analysis centered on 36 scRNA-seq samples, complemented by 14 paired snATAC-seq samples, from which a subset focused on cDCs was used for downstream epigenomic analysis. Out of the 36 scRNA-seq samples, 30 underwent treatment that included ICI therapy, with 19 of them exclusively receiving ICI treatment. Key clinical information, such as progression-free survival (PFS), overall survival (OS), lesional response, treatment details, tissue type, melanoma subtype, sex, age, and time relative to treatment initiation, is visualized in Fig. 1A and listed in Supplementary Data 1.

We conducted data integration for the 36 scRNA-seq samples, implemented dimensionality reduction techniques, and illustrated the 2D cell embeddings. At the highest level, we distinguished tumor, immune, and stromal compartments, along with sample IDs, tissues, melanoma subtypes, sex, treatment groups, treatment state, and lesional responses (Supplementary Fig. 1). We identified 14 cell types for the 189,362 cells, including sample-specific tumor cells (further analyzed in the subsequent section), T lymphocytes (i.e., CD8 T cells, CD4 T cells, other T cells, and cycling T cells), natural killer (NK) cells, B lymphocytes, myeloid cells (encompassing monocytes/macrophages, conventional dendritic cells [cDC], plasmacytoid dendritic cells [pDC], and mast cells), and stromal cells (including endothelial cells, fibroblasts, and keratinocytes; Fig. 1B). Distinct expression patterns of canonical cell type-specific markers are presented in Fig. 1C.

Our single-cell annotations showed highly consistent cell type annotations with previous studies on metastatic melanoma, and contributed more cells than all previous studies (Supplementary Fig. 2A–D)[7–11]. We also confirmed that our cell type markers and cell type annotations remained consistent when jointly analyzing our data with cells from all previous studies combined (Supplementary Fig. 2E).

### Consensus tumor programs and immune cell subtypes in tumor microenvironment
We found that transcriptomic profiles of tumor cells exhibit patient-specific patterns (Supplementary Fig. 1B), as found in previous studies[22]. To correct for these patient specific effects and identify tumor-shared patterns of intra-tumor heterogeneity (ITH), we assigned tumor cells to 15 distinct meta-programs previously compiled in Gavish et al.[22], encompassing 23,997 high-quality cells with a minimum number of 25 expressed genes in the meta-program (Supplementary Figs. 3A, 4A). The predominant gene expression patterns in each meta-program provide insights into the unique functionalities of different tumor cell subpopulations within a tumor. These functionalities span various aspects, including cell-cycle phases (G1/S, G2/M, HMG-rich), upregulation of processes such as epithelial-mesenchymal transition (EMT), hypoxia, interferon and major histocompatibility complex (MHC) class II, MYC, protein maturation, respiration, skin pigmentation, stress, translation initiation, and unfolded protein response (Supplementary Fig. 3B). While most tumor meta-programs exhibited melanocytic differentiation (high *MITF* and *CTNNB1*), two meta-programs-Respiration and Stress-expressed dedifferentiation markers including *AXL, SOX9, EGFR* and *NGFR* (Supplementary Fig. 4B, C). The Respiration program resembles an undifferentiated subtype, while the Stress program reflects a neural crest-like state[22]. Both were later enriched in non-responders, aligning with the known resistance of dedifferentiated melanomas to targeted therapy and immunotherapy[23]. Our analysis confirms previously reported meta-programs[22] and also facilitates correlation of these tumor programs with non-tumor subtypes within the same samples.

For myeloid cells, we classified monocytes into classical (CD14+FCN1+S100A8+S100A9+) and non-classical (CD16+LST1+LILRB2+) monocytes[24]; macrophages into M1, M2 and tumor-associated macrophages (TAMs); and cDCs into cDC type 1 (cDC1), type 2 (cDC2) and a mature and migratory subtype (mregDC; Fig. 2A, Supplementary Fig. 3C). We found that M1 macrophages, which participate in pro-inflammatory responses, are distinguished by their expression of *CXCL9, CXCL10, CD80, FCGR1A*, and *HLA-DR*. In contrast, M2

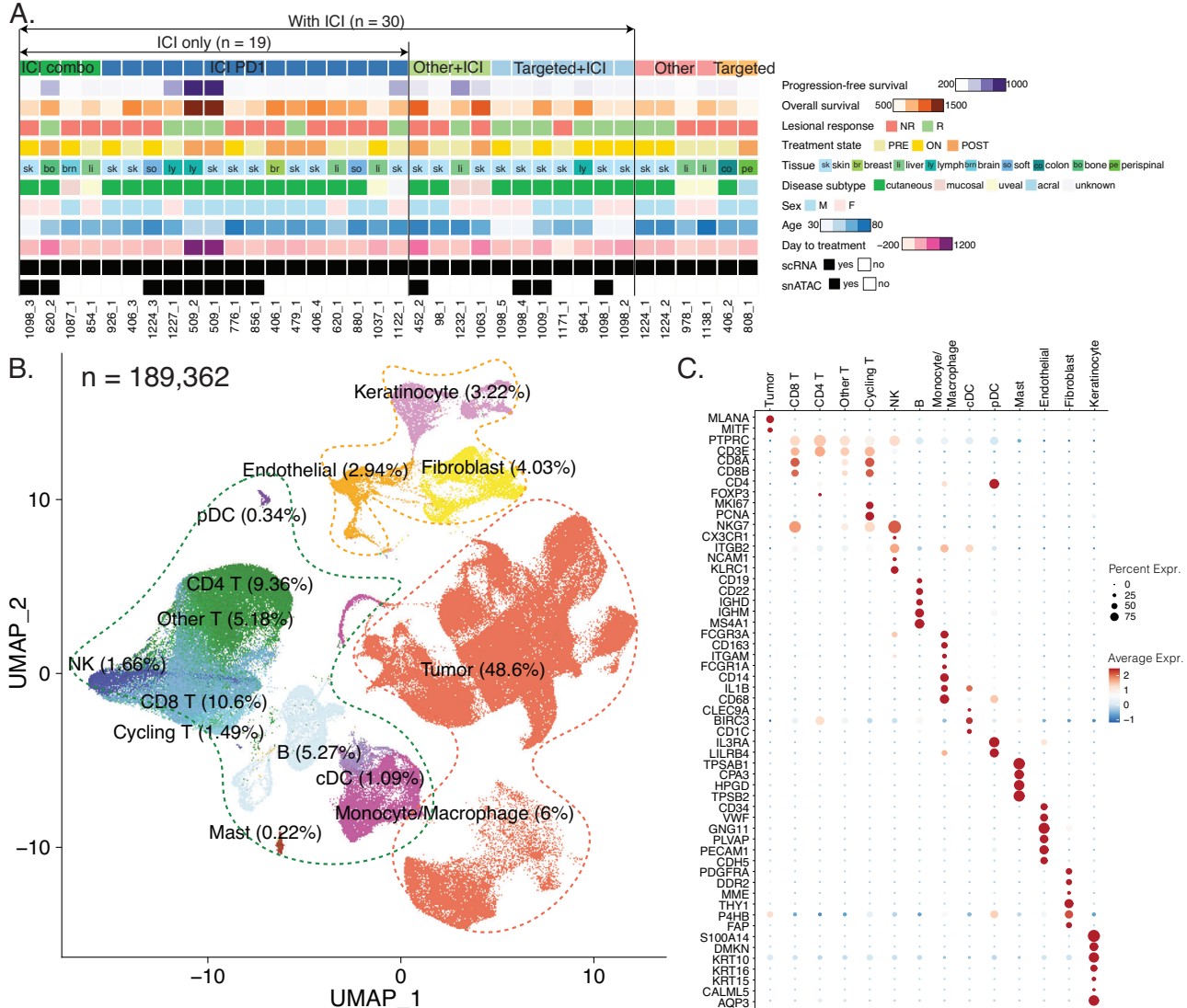

**Fig. 1 | Characterization of cells in human metastatic melanoma tumors.**
**A** Demographic summary of 36 samples included in this study. Pfs, progression-free survival; os, overall survival; NR, non-responder; R, responder. **B** Uniform manifold approximation and projection (UMAP) embedding of 189,362 cells from 36 samples, after quality control, with each color representing a cell type. **C** Dotplot showing average normalized expression and percent normalized expression of marker genes across cell types.

macrophages, known for their immunosuppressive role, exhibited expression of anti-inflammatory molecules *MSR1*, *CD163*, *MRC1*, *C1Q*, *SELENOP*, and *APOE*. TAMs were characterized by their expression of *TREM2*, *ITGAM*, *CCL2O*, and *IL1B*. While we employ the M1/M2 nomenclature to denote inflammatory and anti-inflammatory macrophage states, these clusters align with in vivo macrophage subtypes described in melanoma and other cancers[25]. For example, 'M2-like' macrophages resemble TREM2+ immunosuppressive macrophages, and TAMs share transcriptional overlap with HuTAMs B/C identified in pancreatic cancer[26]. These parallels highlight conserved functional states across malignancies, though evolving macrophage taxonomy underscores the need for standardized, context-specific definitions. Further details about the cDC subtypes will be discussed in a subsequent section.

In the CD8 + T cell compartment, we identified various subtypes, including naive T cells (Tn, CCR7+), effector T cells (Teff, GZMA+GZMB+), memory T cells (Tm, CXCR6+), exhausted T cells (Tex), and natural killer T cells (NKT, CD8+XCL2+; Fig. 2B and Supplementary Fig. 3D). All CD8 + T cell subtypes showed a similar number of UMIs per cell, except for the NKT subtype, which had a higher UMI count as

previously reported (Supplementary Fig. 4D)[27,28]. Among the Teff subset, three clusters exhibited moderate to high expression of cytotoxic markers *GZMA* and *GZMB*. An early activated Teff subgroup was marked by *CD69* expression. Notably, HNRNPH1+ Teff displayed elevated levels of long non-coding RNAs (lncRNA), including *MALAT1*, *NEAT1*, and ribonucleoprotein genes *HNRNPH1* and *HNRNPU*, implicated in regulating T-cell-mediated immune responses, potentially affecting ICI resistance[29]. To confirm that this subtype was not the result of low-quality cells, we scored all CD8 + T cells using a low-quality cell signature[30] and found comparable quality levels across all cells (Supplementary Fig. 4E). Another distinct GZMK+ Teff cluster showed high expression of cytotoxicity-related genes, excluding *CTLA-4*. Within the Tm subset, cells were further categorized into effector memory T (Tem) and exhausted memory T (Texme) based on their expression of cytotoxic genes (*GZMA*, *GZMB*) and the exhaustion marker *TOX*. While checkpoint molecules like *PDCD1*, *LAG3*, *TIGIT*, and *CTLA-4* were broadly expressed, detailed differential analysis revealed three distinct Tex cell clusters, each exhibiting distinct exhaustion and cytotoxicity profiles. Tex/HS displayed high expression of heat shock protein genes and enrichment in heat response-related pathways. The

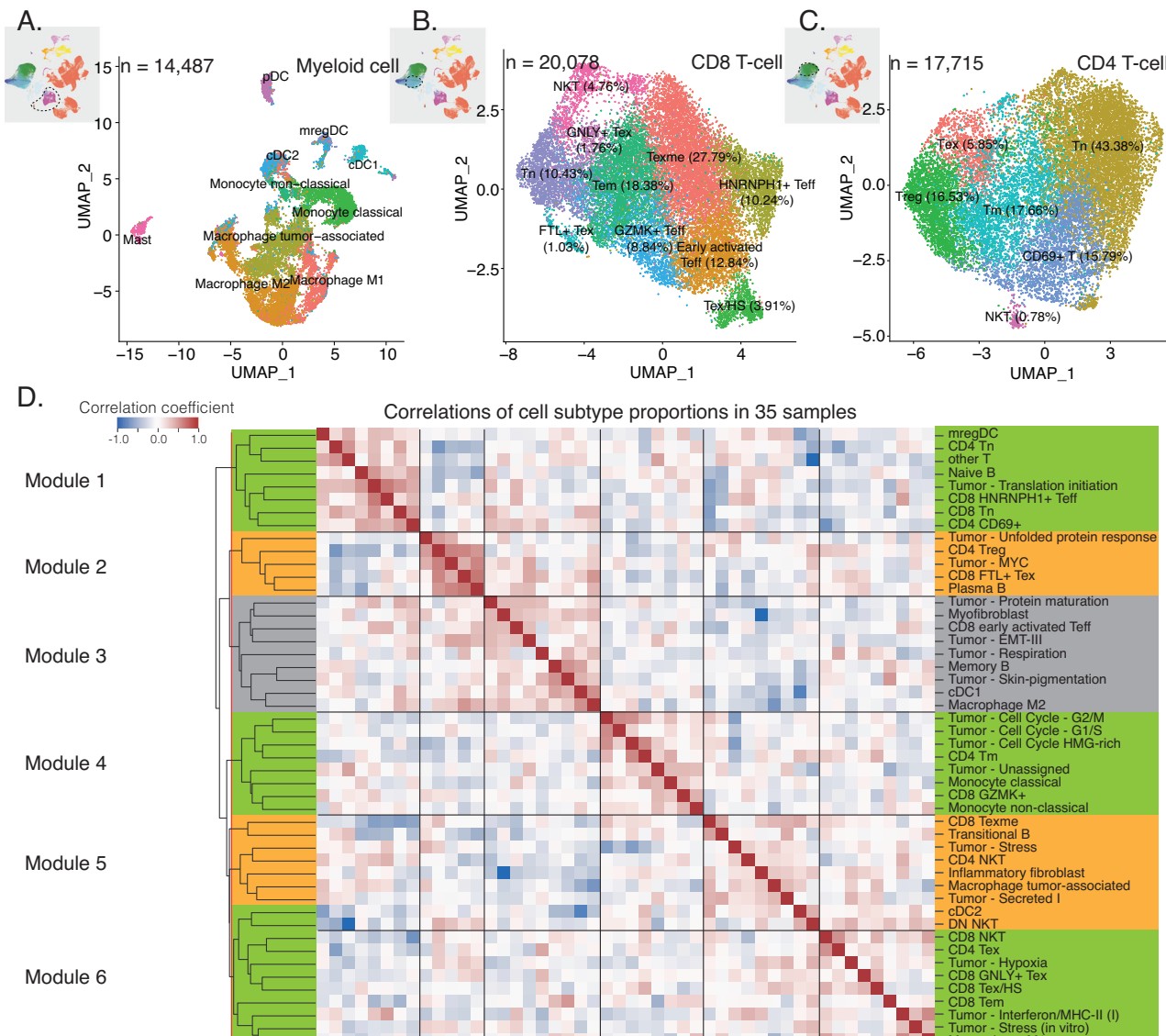

**Fig. 2 | Cell subtypes and their relative proportion correlations in the tumor microenvironment. A–C** UMAP embedding of myeloid cell subtypes of 14,487 cells from 36 samples (**A**), CD8 T-cell subtypes of 20,078 cells from 33 samples (**B**), and CD4 T-cell subtypes of 17,715 cells from 32 samples (**C**). **D** Heatmap illustrating correlation coefficients among the relative proportions of cell subtypes in relation to their corresponding cell types across 35 samples. We identified clusters/modules by segmenting the hierarchical clustering tree at the red line. Source data are provided as a Source Data file.

GNLY+ Tex cluster was unique for expressing the granulysin gene *GNLY* and calcium-binding proteins *S100A10* and *S100A11*. Additionally, the FTL+ Tex cluster exhibited increased expression of *FTL* and *FTH1*, encoding ferritin subunits. The role of ferritin in T cell function is still being explored, but prior studies have linked FTL and FTH1 to immune cell infiltration, particularly in Tregs[31].

We identified six subpopulations of CD4⁺ T cells, including the commonly observed naive T cells (Tn), memory T cells (Tm), regulatory T cells (Tregs), exhausted T cells (Tex), CD69⁺ early activated T cells (CD69⁺ T), and NKT cells (Fig. 2C and Supplementary Fig. 3E). Notably, Tn cells, identified by their high expression of the naive marker *CCR7*, also exhibited significantly elevated expression of ribosomal genes, indicative of cells in a naive or quiescent state[32]. Tm cells expressed the memory marker *CCL5*. Tregs, governed by the transcription factor (TF) FOXP3, demonstrated elevated levels of co-inhibitory molecule *CTLA-4* and *TIGIT* expression. Tex cells exhibited increased expression of checkpoint molecules, including *TOX*, *CTLA-4*, *PDCD1*, and *TIGIT*, along with an increase in glycolytic metabolism,

highlighted by their elevated expression of *GAPDH*. The activated CD69 + T cell phenotype correlated with upregulation of stress-related genes, such as *FOS*, *JUN*, and heat shock protein (HSP) family genes. Additionally, we identified an NKT subpopulation characterized by its unique expression of *XCL1*, *XCL2*, *KLRD1*, and *CD4*, aligning with an innate lymphoid phenotype.

Furthermore, we uncovered a distinct subtype known as double negative NKT cells (DN NKT) among other T cells, characterized by the expression of T cell markers excluding *CD4* and *CD8*, alongside NK markers. DN NKT cells have been proposed to adopt a pro-inflammatory phenotype, contributing to both innate and adaptive immunity[33]. For B cells, we categorized them into four subtypes: naive, transitional, memory, and plasma B cells (Supplementary Fig. 3F). Naive B cells were marked by differentiating markers *IGHD* and canonical markers *FCER2*, *CD19* and *MS4A1*, transitional B cells expressed *CD24* and CD10 (encoded by *MME*)[34], memory B cells were identified by the presence of *CD27* and *CD38*, while plasma B cells expressed markers such as *CD27* at a higher level and *PRDM1*[35]. In addition, we

identified two subtypes of fibroblasts: myofibroblasts, expressing *ACTA2*, *NDUFA4L2*, and *MYL9*, and inflammatory fibroblasts, marked by the presence of *IL6*, *CXCL12*, and *DPT*.

In summary, our analysis revealed a total of 55 distinct cell subtypes within the 14 primary cell types in our cohort (Supplementary Fig. 3G and Supplementary Data 2–4). We later expanded these cell subtype annotations to the integrated single-cell atlas, encompassing data from six studies. This comprehensive subtyping offers an opportunity to explore correlations between these cell subtypes within the TME.

### Cell subtype correlations reveal six modules

To analyze changes in subtype abundance dynamics within the TME, we calculated the relative proportions of cell subtypes in comparison to their respective cell types. Then we conducted correlations between these relative proportions across samples, resulting in the recognition of six distinct modules through unsupervised hierarchical clustering (Fig. 2D, Source Data, Supplementary Data 5).

In Module 1, the mregDC subtype showed strong associations with CD8 Tn, CD8 HNRNPH1+ Teff, CD4 Tn, CD4 CD69 + T cells, and naive B cells (Supplementary Fig. 3H). Module 2 featured correlations among CD8 FTL+ Tex cells, CD4 Treg cells, plasma B cells, and two tumor meta-programs (MYC and unfolded protein response). These associations suggested the suppression of immune functions. Module 3 included cells like cDC1 and M2 macrophages[17]. In Module 4, correlations emerged among CD8 GZMK+ cells, CD4 Tm cells, tumor meta-programs related to the cell cycle, and classical and non-classical monocytes (Supplementary Fig. 3I). These cell subtypes pointed to an effector T cell function. Module 5 included an exhausted T cell subtype, CD8 Texme, and two myeloid cell subtypes, cDC2 and TAMs, showing an immune exhaustion signature (Supplementary Fig. 3J). Lastly, Module 6 consisted of cells involved in antigen presentation and stress response, including tumor meta-programs related to interferon/MHC-II and stress (in vitro), and M1 macrophages (Supplementary Fig. 3K).

Upon focusing on the samples treated with ICI, we identified five out of the six modules as in Fig. 2D, albeit in a different order (Supplementary Fig. 5A). To assess treatment-dependent module dynamics, we performed separate analyses for pre-treatment and on/post-treatment samples. While on/post-treatment samples recapitulated the combined cohort's module patterns, pre-treatment samples exhibited distinct correlations (Supplementary Fig. 5B), revealing that module conservation is context-dependent and shaped by therapy. Expanding our correlation analysis to our integrated single-cell atlas with cell subtype annotations (Supplementary Fig. 6A), we still observed module 1, prompting us to explore its biological significance and its potential associations with clinical variables or responses (Supplementary Fig. 6B). However, the other modules were not conversed across datasets, underscoring the importance of cautious interpretation.

### Mature and migratory cDC abundance associated with ICI response and PFS

To identify cell subtypes with their abundance correlated with lesional response to treatment, we compared cell subtype relative proportions between responding and non-responding tumors. To account for the limited sample size, we combined tumors collected at different treatment timepoints to identify cell types and transcriptomic features consistently enriched in responders and non-responders, irrespective of treatment stage, while acknowledging that treatment itself may influence tumor transcriptome profiles and cell composition (Supplementary Fig. 7A). Using all the samples, we observed more tumor cells with the stress meta-program, more Tregs, and more plasma B cells in non-responders (p < 0.1; Supplementary Figs. 7B, 8A). Tregs and plasma B cells belong to module 2 immune suppression in the

above correlation analysis. The plasma B cells identified in this study expressed high levels of *IGHA1* and *IGHA2* (Supplementary Fig. 3F), suggesting they are likely IgA+ plasma cells. While IgA+ plasma cells have been reported as immunosuppressive in various cancer types[36], their role in the context of ICI therapy remains unclear[35]. We found more inflammatory fibroblasts in responders (p < 0.1, Supplementary Figs. 7B, 8A). When focusing on the samples treated with only ICI (combo or anti-PD1), we detected more tumor cells with the respiration tumor meta-program and M2 macrophages in non-responders, and more classical monocytes and mregDCs in responders (p < 0.1, Fig. 3A). This finding highlights these cell types in mediating ICI-specific effects, and shows consistency with the known role of M2 macrophages and classical monocytes mediating anti- and pro-ICI effects, respectively[37]. mregDCs, transcriptionally characterized by single-cell studies[19,38,39], were reported to be more abundant in responsive triple-negative breast cancer patients during anti-PD-1 treatment[40]. PD-L1-expressing DCs (mregDCs highly expressing PD-L1) were shown as key mediators of ICI effects in mouse models[41]. However, the underlying mechanism remains incompletely understood. Interestingly, neither cDC1 nor cDC2 was associated with treatment response in all the samples or subsets of samples (Supplementary Figs. 8B, 9A). mregDC did not show differential abundance in samples treated with therapies other than anti-PD1 only (Supplementary Fig. 9B). Relative proportions of cDC in all cells, all immune cells or all mononuclear phagocytes were not associated with treatment response (Supplementary Fig. 9C). In summary, these findings suggest that the relative proportion of cDCs showing the mature and regulatory phenotype could affect tumor's response to ICI.

To find a more general trend rather than examining individual mregDC proportions, we defined the mregDC high samples as having at least 20% mregDC in cDC, which roughly contained 25% of all the samples (n = 10, total = 35). We observed significant PFS benefit and marginally significant OS benefit for ICI-treated samples with high relative proportions of mregDCs (Fig. 3B). Using the same threshold to group samples into mregDC high versus low, we did not observe significant survival benefit for samples with other treatments, although mregDC high samples always showed higher survival probability compared to mregDC low samples (Supplementary Fig. 7C, D). We conducted multivariate Cox regression analysis to address dataset heterogeneity and confirmed that both continuous and binarized mregDC proportions are significant predictors of PFS in ICI-treated samples. To account for non-independence of longitudinal samples from the same patient, survival analyses (Kaplan-Meier and Cox regression) were revised to include only the latest timepoint per patient, reducing the cohort to n = 11 samples. Despite this adjustment, trends in survival benefit for mregDC-high tumors persisted (Supplementary Fig. 9D), underscoring the robustness of our findings. None of the other cell subtypes associated with ICI response showed stratification of patient's survival; however, their survival curve splits were consistent with their corresponding effect directions for treatment response (Supplementary Fig. 9E). A previous single-cell metastatic melanoma study established an association between TCF7 + CD8 T cells and ICI response[9]. Here, we show that mregDC relative proportion is in fact particularly correlated with TCF7+ versus TCF7- CD8 T cell ratio (Supplementary Fig. 7E) and confirmed more TCF7 + CD8 T cells in responsive ICI-treated samples (Supplementary Fig. 7F). We also detected significant PFS and marginal OS benefit in ICI-treated samples with a higher TCF7+ ratio (Supplementary Fig. 7G). Similar to the mregDC relative proportion, the TCF7 ratio did not stratify patient's survival in samples with other treatments (Supplementary Fig. 9F). However, when we combined these two potential biomarkers, they showed significant survival prediction power in all samples: samples with high mregDC proportion and high TCF7 + CD8 ratio had higher survival probability regardless of treatment type compared to samples with either low mregDC proportion or low TCF7 + CD8 ratio

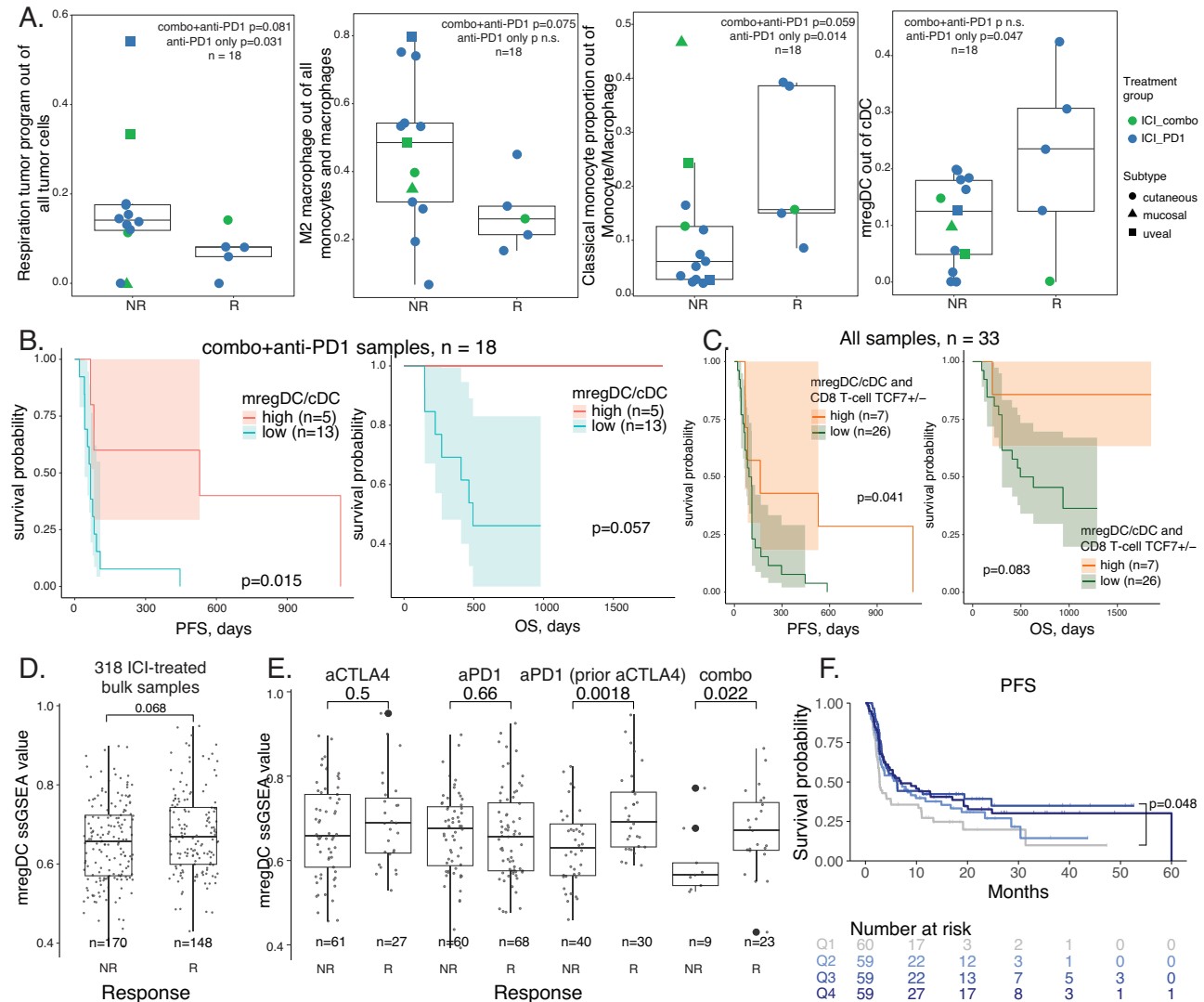

**Fig. 3 | Mature and regulatory cDCs associated with ICI response and patient survival. A** Boxplots comparing relative proportions of four subtypes between ICI non-responders ($n = 13$) and responders ($n = 5$). Each dot represents a sample, with its color corresponding to the type of ICI and its shape corresponding to the melanoma subtype. One sample lacking detectable tumor cells, monocytes/macrophages and cDCs was excluded from this analysis and the following analysis. **B** Survival plots for 18 samples only treated with ICI and split by 20% of mregDC relative proportion, which roughly contained 25% of all the samples. **C** Survival plots for 33 samples and split by both mregDC relative proportion and TCF7 + CD8 T ratio. Three samples lacked CD8 + T cells and were excluded from this analysis. **D** Boxplot comparing mregDC signature scores, calculated using ssGSEA, between non-responders and responders in 318 ICI-treated bulk RNA-seq samples. **E** Boxplot

comparing mregDC signature scores, calculated using ssGSEA, between non-responders and responders split by four treatment types for 318 samples. **F** Progression-free survival plot for the 318 bulk RNA-seq samples split by their mregDC scores. *P*-values for boxplots in all panels were calculated using the two-sided Wilcoxon rank sum test. *P*-values for survival plots in all panels were calculated using the Log-rank sum test. NR, non-responder; R, responder; combo, anti-PD-1+anti-CTLA-4; PFS, progression-free survival; OS, overall survival. The boxplots in panels **A**, **D** and **E** show the distribution of the data, with the central line representing the median (50th percentile), the box indicating the interquartile range (IQR) from the 25th to 75th percentile, and the whiskers extending to the minimum and maximum values within 1.5 times the IQR. Outliers beyond this range are plotted as individual points. Source data are provided as a Source Data file.

(Fig. 3C). Interestingly, no samples had a high mregDC/cDC proportion and low TCF7 + /− ratio.

Moreover, we tested a derived mregDC signature with 886 genes (average log2 fold change > 0, adjusted *P*-value < 0.05; Supplementary Data 3) in a meta-metastatic melanoma cohort treated with ICI and with tissue-level RNA-seq data available[42]. We validated the association between mregDC signature scores and patient's response to ICI ($n = 318$, Fig. 3). Because our validation cohort included patients treated with anti-CTLA monotherapy, anti-PD1 monotherapy, anti-PD1 with prior anti-CTLA4, and combination therapy, we further showed that this effect persisted in both anti-PD1 with prior anti-CTLA4 and combination therapy subgroups (Fig. 3E), suggesting that mregDC's predictive capacity is regimen-specific. The same mregDC signature score

split into four quartiles showed significant PFS difference and marginal OS difference in this meta-cohort (Fig. 3F and Supplementary Fig. 7H). While the mregDC signature correlated with overall immune infiltration (ESTIMATE immune score[43], $p < 2.2 \times 10^{-16}$; Supplementary Fig. 7I), multivariate analysis revealed that the combination of mregDC and immune score significantly predicted PFS ($p = 0.047$), though neither covariate reached individual significance. This suggests mregDC and immune infiltration capture complementary aspects of the TME, jointly enhancing prognostic power beyond either metric alone.

## Transcriptional landscape of mature and migratory cDCs
We found that cDCs clustered into three subtypes: cDC1, cDC2 and mregDC (Fig. 4A and Supplementary Fig. 10A), and sought insights in

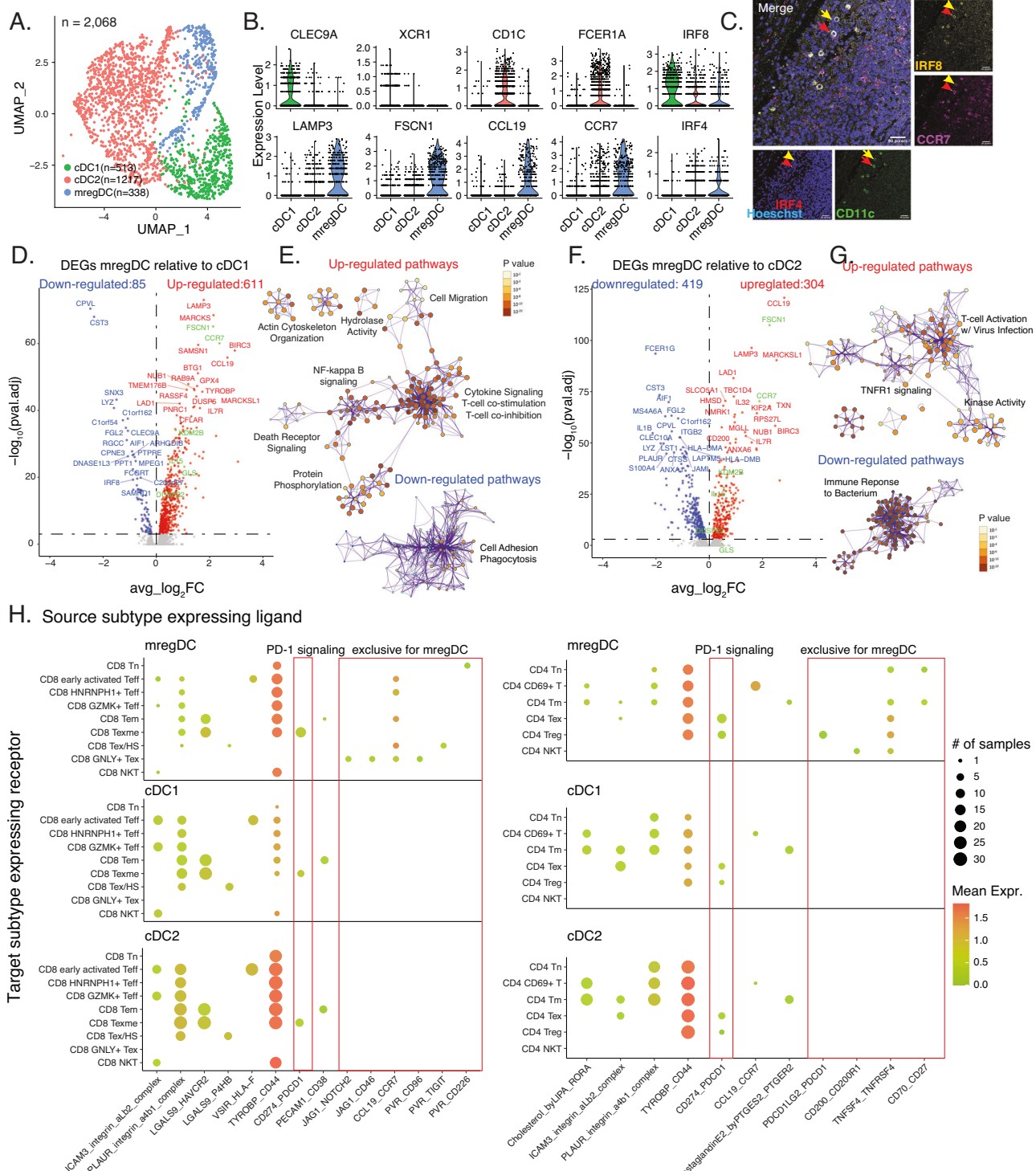

**Fig. 4 | Transcriptional landscape and interactome of cDC subtypes. A** UMAP embedding of 2,068 cDCs from 35 samples colored by the subtypes. **B** Violin plots of canonical marker genes for cDC1, cDC2 and mregDC. **C** Multiplex immunofluorescence overlay of a human sentinel lymph node tissue (number of markers = 4). Representative images of mregDC marker CCR7 (pseudocolored magenta), dendritic cell marker CD11c (pseudocolored green), IRF8 and IRF4 marker (pseudocolored yellow and red, respectively) are shown on the right. CD11c + CCR7+ cells expressing either both IRF4+ and IRF8+ markers (red arrow) or IRF8+ alone (yellow arrow) cells are indicated on each of the panel figures. Nuclei were counterstained with Hoechst 33342. The immunofluorescence results are representative of two independent tissue samples. Scale bar = 8 μm. **D, F** Volcano plots showing differentially expressed genes (DEGs) between 338 mregDCs and 513

cDC1s (**D**) and between 338 mregDCs and 1217 cDC2s (**F**) by using cDCs from 35 samples. An adjusted *P*-value of 0.001 was used to call significant DEGs. Examples of genes with mregDC-specific enhancers were colored in green, which were mentioned in Fig. 5. **E, G** Networks of enriched terms for DEGs up- or down-regulated in mregDC compared to cDC1 (**E**) or cDC2 (**G**). **H** Dotplots illustrating inferred cell-cell communications by CellPhoneDB[57], either with differential activities between mregDC and other cDC or exclusively detected in mregDC. The left and right panels plotted interactions with cDC expressing ligands and CD8 or CD4 T-cell expressing receptors, respectively. *P*-values for volcano plots and dotplots were calculated using the two-sided Wilcoxon rank sum test. The hypergeometric test was used for panels E and G without multiple comparison adjustments.

their respective functions. cDC1 was characterized by high expression of canonical markers *CLEC9A* and *XCR1*, while cDC2 displayed elevated levels of *CD1C* and *FCER1A*. mregDC exhibited elevated expression of *LAMP3*, *FSCN1*, *CCL19*, and *CCR7*, consistent with prior findings[38,39]. Notably, cDC1 showed significant expression of its lineage-specific TF *IRF8* among the three subtypes[44]. However, *IRF4*, the cDC2 lineage-specific TF, is expressed most prominently in mregDC compared to cDC1 and cDC2 (Fig. 4B). Multiplex immunofluorescence staining of sentinel lymph node tissues from metastatic melanoma patients confirmed protein-level co-expression of IRF4 with IRF8 in CD11C + CCR7+ mregDCs (Fig. 4C). cDC1, cDC2 and mregDC scores based on previously reported signatures[19] confirmed our classification of the three subtypes (Supplementary Fig. 10B). The majority of published mregDC signatures are confirmed, with exceptions in Th2 response genes (Supplementary Fig. 11A, B)[19]. Similar to mouse mregDCs[45], human mregDCs express elevated levels of *IL4I1*, *CCL17*, *CCL22*, *TNFRSF4*, and *BCL2L1* compared to human cDC1 and cDC2 subsets. However, human cDC2s uniquely exhibit higher expression of *STAT6* compared to both human cDC1s and mregDCs (Supplementary Fig. 11B).

Differential expression analysis revealed distinct gene expression profiles and pathways between mregDC and cDC1 as well as cDC2. Specifically, mregDC exhibited a higher number of up-regulated genes than down-regulated genes compared to cDC1 (611 vs. 85, Fig. 4D and Supplementary Data 6). The up-regulated genes were enriched in pathways involved in actin cytoskeleton organization, hydrolase activity, cell migration, NF-κB signaling, death receptor signaling, protein phosphorylation, cytokine signaling, T-cell costimulation and T-cell co-inhibition, aligning with the mature, activated, migratory and immunoregulatory features of mregDC (Fig. 4E and Supplementary Data 7). When comparing mregDC with cDC2, mregDC showed higher expression of 304 DEGs and lower expression of 419 genes (Fig. 4F). The up-regulated pathways predominantly related to T-cell activation with virus infection, TNFR1 signaling and kinase activity, whereas the down-regulated pathways included immune response to bacterium (Fig. 4G). Based on the observed differences in pathways between mregDC and cDC1/cDC2, mregDC likely derives from cDC1, with gene programs associated with relevant phenotypes activated.

Upon comparing the expression patterns of select genes from a prior study[46], we noted that many markers previously associated with high expression in cDC1 populations were exclusively expressed by mregDCs in our cohort. These markers include *CCR7* and *CCL22* for migration, *CD86*, *CD80*, and *CD40* for maturation, *PDL1*, *PDL2* and *RANK* for immunoregulation, *TAPBP* for cross-presentation, and *CSF2RA* and *CSF2RB* for the granulocyte-macrophage colony-stimulating factor receptor (GM-CSFR). This finding further suggests that cDC1 cells could be a primary source of mregDCs in the human TME (Supplementary Fig. 11C). Additionally, there are markers equally expressed by cDC1s and mregDCs, such as *IL12B* and *FLT3*. Furthermore, we compared MHC I and MHC II gene expression across the three cDC populations and found that MHC II genes were highest in cDC2, while MHC I genes were upregulated in mregDC, suggesting mregDCs primarily present antigens to CD8 + T cells via MHC I (Supplementary Fig. 11D).

To further understand differences between mregDCs and cDC1/cDC2, we performed trajectory analysis[47], regulon (TF and its targets) inference[48], and immune response enrichment[49] inference for the three subtypes. Trajectory analysis revealed that both cDC1 and cDC2 possess the potential to transition into mregDCs (Supplementary Fig. 10C), consistent with findings from experimental studies using mouse cDCs[19,45]. Despite its lower expression in mregDC compared to cDC1, IRF8 exhibited the highest regulon specificity score in mregDC, indicating its potential role in mediating the transition from cDC1 to mregDC (Supplementary Fig. 10D). Other top regulons included those associated with NF-κB signaling (REL, NFKB1, NFKB2), AP-1 signaling (FOSB, FOSL2), and oxidative stress response (MAFG, NFE2L2). IRF1,

another regulon enriched in mregDC, has been implicated in driving antitumor immunity in mouse models through its control by NF-κB signaling[50]. Additionally, IKZF1, another enriched regulon in mregDC, is known to regulate DC function in humans[51]. KLF3 and KLF6 were also among the top enriched regulons in mregDC. Notably, KLF4, a member of the same Krüppel-like factor family, has been reported to promote Th2 cell responses in IRF4-expressing cDCs[52]. Immune response enrichment[49] indicated that mregDCs exhibited transcriptional responses to IL1β, TNFα, IL12, IFNβ, GM-CSF, TSLP, and IL18, while cDC1 showed enrichment for Leptin, Flt3l, LIF, and IL21 (Supplementary Fig. 10E). GM-CSF[53] and TSLP[54] are known to promote cDC maturation and activation, while Flt3l is crucial for cDC precursor development and cDC maintenance[55], and IL21 was shown to inhibit DC activation and maturation[56]. Pro-inflammatory cytokines such as IL1β, TNFα, IL12, and IFNβ appeared to influence mregDCs as well.

## Differential interactome between mature and migratory cDCs and other cDCs

To examine unique cellular interactions between the three cDC subtypes and other cell populations, we conducted cell-cell communication analysis utilizing ligand-receptor co-expression patterns[57], revealing immune-stimulatory and regulatory roles of mregDCs in the TME, and the survival and activation signals they receive from T cells and other myeloid cells. We annotated interactions with varying inferred activities in mregDC compared to other cDCs or those exclusively present in mregDCs (Supplementary Data 8, 9). Notably, CD8 T-cells, CD4 T-cells, and monocytes/macrophages exhibited the highest number of such interactions with cDCs, either as sources or targets (Supplementary Fig. 10F). For CD8 T-cells, mregDC exhibited heightened interactions with CD8 exhausted memory T cells (CD8 Texme), exhausted CD4 T cells (CD4 Tex), regulatory T cells (Treg), and M1 macrophages through CD274 (PD-L1) and PDCD1 (PD-1) interactions (Fig. 4H and Supplementary Fig. 11E). This finding underscores the significance of these interactions in mediating ICI response, particularly anti-PD-1 treatment. Specific molecular interactions were exclusive to mregDCs, including interaction between the up-regulated CCL19 cytokine in mregDCs and CCR7-expressing CD8 effector and exhausted T cells, as well as early activated CD4 T cells (CD4 CD69 + T), suggesting potential recruitment of these T cells by mregDCs. Additionally, co-immunoregulatory interactions such as TIGIT-PVR, PDCD1LG2 (PD-L2)-PDCD1 (PD-1), and CD200-CD200R1 were observed between mregDCs and exhausted CD8 T cells with high heat shock gene expression (CD8 Tex/HS), Tregs, and CD4 NKT cells (Fig. 4H). The interaction involving PD-L2 and PD-1 could represent another site of action with anti-PD-1 treatment. Apart from co-immunoregulatory signals, we identified interactions between CD70 and CD27 on mregDCs and naive and memory CD4 T cells, potentially mediating the co-occurrence of mregDCs and naive CD4 T cells (Fig. 2F). We also observed decreased levels of prostaglandin E2 interaction with PTGER2 on memory CD4 T cells (Fig. 4H), suggesting inhibition of Th2 differentiation for these cells[58], aligning with the lower levels of Th2 response genes seen in mregDCs (Supplementary Fig. 11B). Additionally, mregDC also up-regulated the interaction of RARRES2 (chemerin) and CMKLR1 with non-classical monocytes (Supplementary Fig. 11E), and indeed chemerin has been reported to act as a tumor suppressive cytokine in mouse melanoma models by recruiting innate immune cells into the TME[59].

Conversely, TNFSF9 (4-1BBL) and TNFRSF9 (4-1BB) exhibited co-expression between CD8 GNLY+ Tex and mregDC (Supplementary Fig. 10G). Another ligand-receptor interaction notably up-regulated between mregDC and all CD8, CD4, monocyte and macrophage subtypes except CD8 Tex/HS was TNFRSF11B and TNFSF10 (TRAIL; Supplementary Figs. 10G, 11E). TNFRSF11B is known to compete for TRAIL binding to death-activated receptors, offering a mechanism to prevent apoptosis in the presence of TRAIL[60]. Given the previously observed

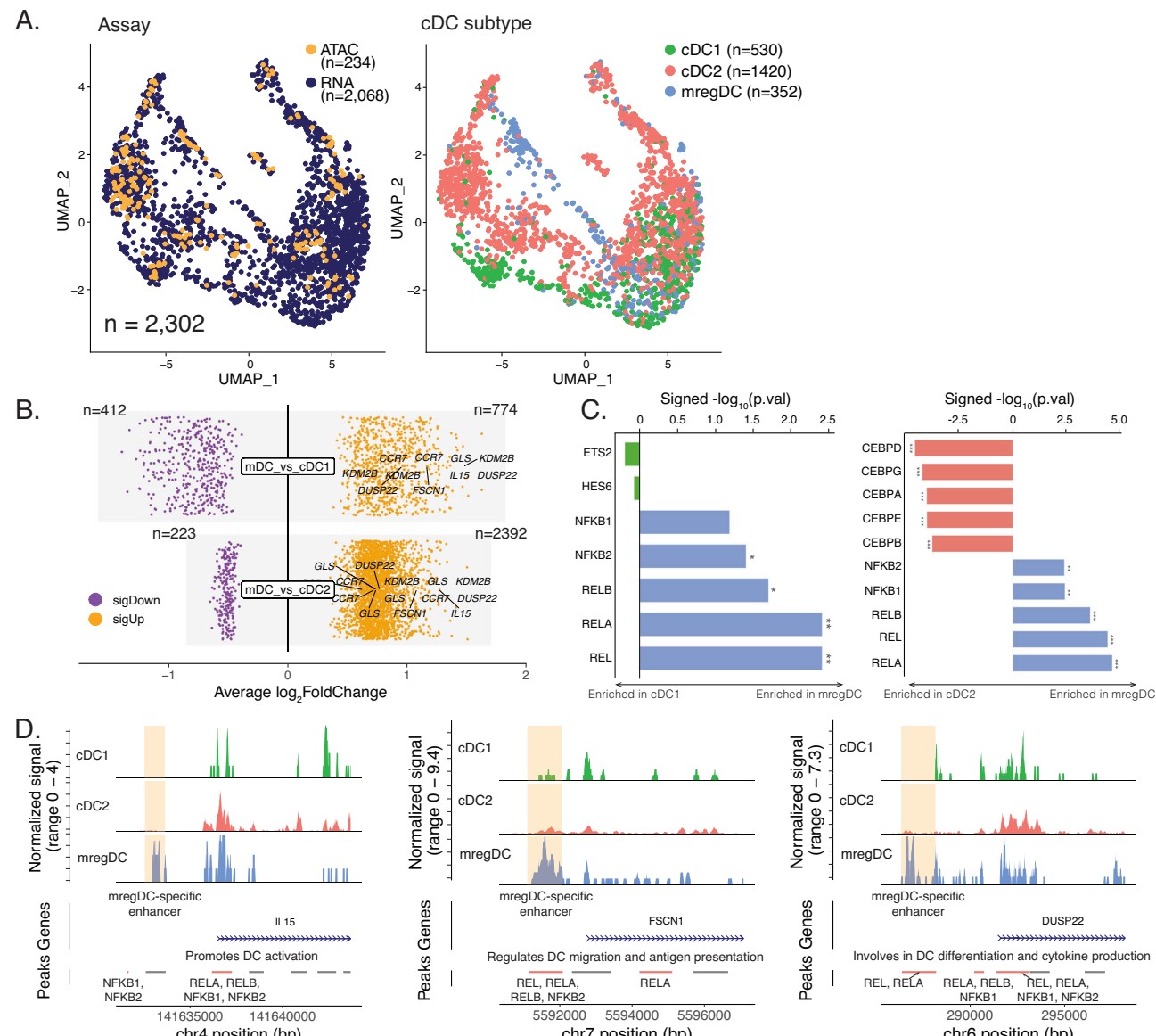

**Fig. 5 | Epigenomic landscape of cDC subtypes using snATAC-seq data. A** Co-embedding of 2,302 cDCs from 35 samples with scRNA-seq data and 14 samples with snATAC-seq data, with colors corresponding to assays (left) or subtypes (right). **B** Differentially accessible regions between 14 mregDCs and 17 cDC1s (top), and 14 mregDCs and 203 cDC2s (bottom). The *p*-value cutoff is 0.01. **C** Motifs enriched in differentially accessible regions between 14 mregDCs and 17 cDC1s (left), and 14 mregDCs and 203 cDC2s (right). Significantly enriched motifs were marked with an asterisk (adjusted *p*-value < = 0.05). **D** Track plots comparing normalized number of reads across cDC subtypes (17 cDC1s, 203 cDC2s and 14 mregDCs) underlying enhancers associated with genes important for cDC functions. mDC is short for mregDC in some figure panels due to space constraints. A peak is colored orange if it contains motifs for members of the NF-κB transcription factor family. The same data object was used for all panels. The two-sided Wilcoxon rank sum test was performed in panel B without multiple comparison adjustments. The hypergeometric test was used for panel C without multiple comparison adjustments.

up-regulation of death receptor signaling in mregDCs, the heightened expression of TNFRSF11B could signify another pro-survival strategy for mregDC. Simultaneously, TNFSF12 (TWEAK) expressed by CD4 Tregs likely interacted with TNFRSF25 on mregDCs (Supplementary Fig. 10G), which was suggested to modulate the innate response and its transition to adaptive Th1 immunity[61]. We also observed an up-regulated co-expression of NRP2 (neuropilin-2) and SEMA3C between classical monocytes and mregDCs (Supplementary Fig. 11E). Neuropilin-2 is known to play an essential role in the activation of DCs[62].

### Epigenomic landscape of mature and migratory cDCs
For the subset of 14 samples with snATAC-seq data, we performed label transfer[63] from the scRNA-seq atlas to annotate major cell types in the epigenomic space. Given the significance of mregDC in mediating ICI response, we isolated cDCs from both scRNA-seq and snATAC-seq data, integrated them into a unified UMAP space, and identified cDC subtypes through unbiased clustering and RNA-based annotations (Fig. 5A). Using differential accessibility analysis, we found distinct regions between mregDC and cDC1, and mregDC and cDC2, with more than double the number of differentially accessible regions (DARs) observed between mregDC and cDC2 compared to mregDC and cDC1 (2615 vs. 1186; Fig. 5B and Supplementary Data 10). For accessible regions with specific activities in mregDC, cDC1 showed a higher global activity correlation across these peaks than cDC2 (Supplementary Fig. 12A). Despite the limited number of cDC1 and mregDC cells in the ATAC data, our results suggest a more pronounced epigenomic change between mregDC and cDC2. Correspondingly, DARs between

mregDC and cDC2 are significantly enriched in more biological processes compared to DARs between mregDC and cDC1 (Supplementary Fig. 12B, C and Supplementary Data 11), consistent with the diverse biological pathways enriched in DEGs between mregDC and cDC2 observed in our RNA data (Fig. 4D, F). The DARs with higher activities in mregDC compared to cDC1 or cDC2 were enriched in motifs for all five members of the NF-κB transcription factor family, including RELA, RELB, REL, NFKB1 and NFKB2 (Fig. 5C). This enrichment was validated by the previous SCENIC analysis, which identified upregulated NF-κB regulons driving transcription (Supplementary Fig. 10D). Although we did not find motifs significantly enriched in DARs with higher activities in cDC1, motifs for the C/EBP family of TFs were significantly enriched in DARs showing higher activities in cDC2 (Fig. 5C). Notably, enhancer accessibility for *IL15*, *FSCN1*, and *DUSP22*−genes crucial for DC functions−were identified only in mregDCs (Fig. 5D), indicating potential epigenomic reprogramming during cDC maturation and activation. Both *FSCN1* and *DUSP22* enhancers contain motifs for the NF-κB TF family, highlighting the important role of NF-κB signaling in mediating DC maturation. Additionally, we pinpointed enhancers for *KDM2B*, implicated in regulating IL6 expression[64]; *GLS*, glutaminase 1, a prognostic biomarker linked to DCs and immunotherapy response in breast cancer[65]; and *CCR7*, pivotal for DC migration and lymph node homing (Supplementary Fig. 12D). These genes (*IL15*, *FSCN1*, *DUSP22*, *KDM2B*, *GLS*, and *CCR7*) are significantly upregulated in mregDCs compared to cDC1s and cDC2s (Fig. 4D, F), indicating epigenetically driven transcriptional regulation.

## Molecular and cellular factors associated with mature and migratory cDC proportions

Having established an association between mregDC proportion and ICI response, and its predictive value for patients' PFS, with the recognition of mregDC as a distinct cDC subtype with unique transcriptional and epigenomic signatures, we next shifted our focus to understanding the molecular and cellular determinants influencing mregDC proportions within the TME. To achieve this, we stratified the cohort into two groups based on mregDC levels: high vs. low, using the previously established threshold from the survival analysis. For each cell type and subtype, we identified cell states associated with mregDC proportion and DEGs characterizing the mregDC-associated cell state, while adjusting for sample- and cell-level covariates (Fig. 6A). Subsequently, we investigated the enriched biological pathways in the DEGs and the differential cell-cell interactions between high and low mregDC samples using logistic regression models, accounting for sample-level covariates.

At the cell type level, tumor cells, fibroblasts, and B cells exhibited the highest number of DEGs linked to mregDC proportions (Fig. 6B). Due to cell subtype heterogeneity, DEGs identified at the cell type level may reflect varying proportions of cell subtypes within the cell type across the two groups. Indeed, we observed this for B cells, where the proportion of naive B cells correlated with mregDC proportions, and naive B cells were the predominant subtype within B cells in terms of abundance (Fig. 2D). However, for tumor cells and fibroblasts, a significant number of DEGs persisted at the cell subtype level, indicating that subtype-specific DEGs primarily drove the cell type-level differences (Supplementary Fig. 13A). Interestingly, for the skin pigmentation tumor program, correlated with mregDC proportion (Fig. 2F), we identified only 6 DEGs (Supplementary Fig. 13A), highlighting the model's specificity.

Analyzing the top DEGs for each cell type, we observed upregulation of CCL22 in cDCs among mregDC high samples, consistent with its role as an mregDC marker (Fig. 6C and Supplementary Data 12). At the cell subtype level, MHC class II genes were upregulated in mregDC high samples across various tumor programs, including HLA-DRB1, HLA-DPA1, HLA-DPB1 and HLA-DMA in the respiration program, HLA-DMA in cell cycle G2/M program, HLA-DPA1 and HLA-DRA in cell cycle

HMG-rich program, and HLA-DRA and HLA-DPA1 in the stress program (Supplementary Fig. 13B and Supplementary Data 13), indicating the immunogenic potential of these tumor programs in mregDC high samples. Notably, melanoma dedifferentiation marker NGFR and drug resistance marker AXL were upregulated in the tumor cell cycle G2/M program, suggesting potential interactions with mregDC that remain to be elucidated. In fibroblasts, inflammatory fibroblasts exhibited upregulation of CCL21, while myofibroblasts upregulated IL24 and CXCL10 in mregDC high samples (Supplementary Fig. 13C and Supplementary Data 13). These cytokines and chemokines possess proinflammatory and anti-tumor properties, indicating an anti-tumor microenvironment in mregDC high samples.

These cell type-specific DEGs showed enrichment in upregulated ECM pathways in tumor cells and fibroblasts, and downregulated cell cycle pathways across various cell types including tumor, CD4 T, and cDC cells (Fig. 6D). Using prior spatial transcriptomics data from a tumor slide[11], we demonstrated colocalization of myeloid cells with a high mregDC signature and fibroblasts enriched for ECM pathway activity (Supplementary Fig. 13D, E). Additionally, fibroblasts exhibited downregulation in muscle contraction-related pathways and the VEGFα VEGFR2 signaling pathway, indicating a shift from myofibroblasts to inflammatory fibroblasts in mregDC high samples. Subtype-level analysis revealed upregulation of the ECM pathway in the tumor translation initiation and unfolded protein response programs (Supplementary Fig. 13F). The tumor respiration program displayed upregulation in PD-1 signaling and inflammation pathways, suggesting its involvement in anti-PD-1 response and its association with mregDCs. Furthermore, cDC and TAM showed specific cell-cell communications with tumor, T and NK cells in mregDC high samples (Supplementary Fig. 14A, B and Supplementary Data 14).

## Discussion

We compiled a single-cell atlas of metastatic melanoma from 15 responding and 21 non-responding tumors, comprising over 180,000 cells across 55 annotated subtypes. Using this atlas, we identified six cellular programs with correlated subtype proportions, indicating regulatory relationships within the TME. We validated one of the six cellular programs by integration with five previously published single-cell studies[7-11]. Notably, we observed a higher relative proportion of mregDCs in responders compared to non-responders to ICI treatment. This observation was confirmed in an independent ICI-treated bulk meta-cohort. When combining the mregDC proportion with the TCF7 +/− CD8 T cell ratio, we stratified patients' survival across treatments. Furthermore, our characterization of mregDCs using scRNA-seq and snATAC-seq data indicates their potential origin from cDC1, driven by intrinsic TFs like IRF8, and extrinsic factors such as pro-inflammatory cytokines (IL1β, TNFα, IL12, etc.), along with ECM genes from specific tumor cells and fibroblasts. These factors potentially induce epigenomic reprogramming of naive cDCs, leading to the adoption of new phenotypes and functions.

We identified specific cellular ligand-receptor co-expressions between mregDCs and T cells and other myeloid cells, potentially explaining ICI response in mregDC high samples. Our analysis revealed that mregDC proportions correlated with naive CD8 T, naive CD4 T, and naive B cell proportions in the TME. Its significant correlation with overall immune infiltration as inferred by the ESTIMATE immune score in a meta-RNA-seq cohort aligns with clinical evidence that DC recruitment and activation drives immune infiltration (NCT01976585)[66], suggesting mregDCs may orchestrate broader anti-tumor immunity rather than merely reflecting it. Prognostic power of the derived mregDC signature supports its role as a biomarker of functional DC engagement, which could guide strategies to amplify DC-mediated immune infiltration. At a molecular level, we predicted mregDCs interacting with naive CD4 T cells via CD70 and CD27. CD27, a T cell costimulatory molecule, is known to

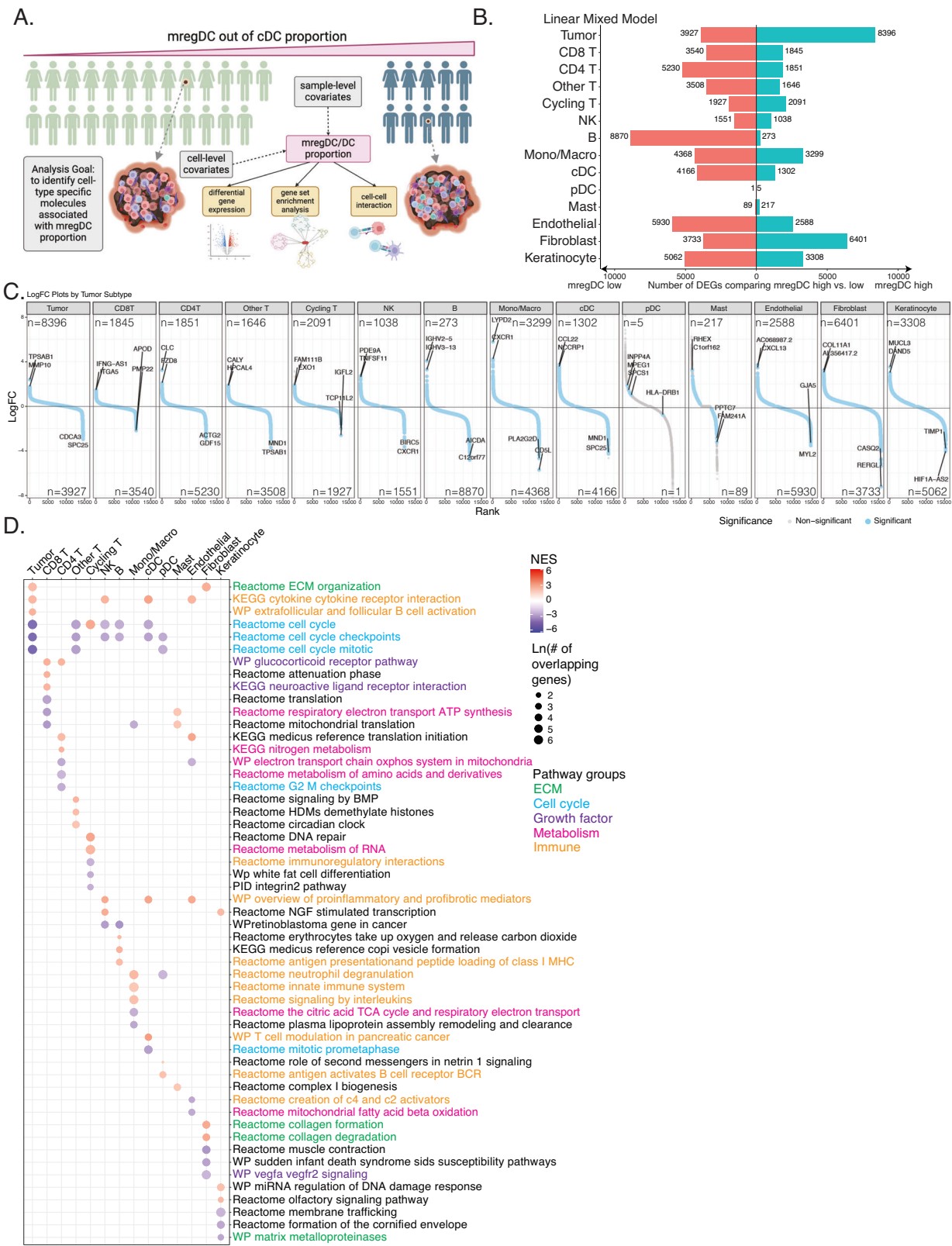

support antigen-specific expansion of naive T cells[67], while CD70 expressed on cDCs has been shown to promote the development of CD4 T cells producing various effector cytokines[68]. Additionally, the interaction between CD27 on naive CD8 + T cells and CD70 on APCs has been recently shown to determine T cell memory fate, with CAR-T cells generated with CD27 costimulation showing superior anti-tumor efficacy[69]. This molecular interaction underscores the intimate relationship between mregDCs and naive T cells, providing additional DC-T targets beyond CD40-CD40L. However, the interpretation of these costimulatory signals needs to consider regulatory signals such as PD-1/PD-L1 and IDO1, which have been shown to affect the proliferation and effectiveness of naive CD8 T cells and induce the differentiation of naive CD4 T cells into Tregs using in vitro co-culture assays[70,71].

**Fig. 6 | Cell type-specific molecular factors associated with mregDC proportions. A** Illustration of sample splitting, model used, and downstream analyses performed to identify cell type-specific molecules associated with mregDC proportion. This analysis included 25 mregDC low samples and 10 mregDC high samples. Created in BioRender. Yang, J. (2025) https://BioRender.com/fdiieqb. **B** Bar plots showing the number of cell type-specific differentially expressed genes (DEGs) significantly upregulated or downregulated in 10 mregDC high samples compared to 25 mregDC low samples, detected by a linear mixed model. **C** Aggregated rank-logFC plots across cell types showing DEGs passing the adjusted

*P*-value threshold 0.05 in light blue and the rest of the genes in gray. The top two genes with large average log2 fold changes in both directions are labeled. **D** Dot plots illustrating the top three pathways enriched by the up and downregulated DEGs for each cell type. The pathway names were colored by biological themes. The same 35 samples were used for all panels. For panel C, negative binomial mixed models were fit to the data, with two-sided *t* values used to determine statistical significance. Multiple comparison adjustments were carried out for the obtained *p*-values. Please see the Methods section for more details.

The higher proportion of mregDCs observed in ICI-responding tumors, both in the single-cell dataset and a large ICI-treated bulk meta-cohort, is consistent with the expression of a direct target of anti-PD-1 treatment on their surface. Studies in mouse models have shown that PD-L1 on cDC1s attenuates T cell activation and regulates response to ICI[41]. Additionally, in a cohort of pembrolizumab-treated breast cancer patients, the relative frequency of mregDCs positively correlated with T-cell expansion following anti-PD-1 treatment, and mregDCs supported T-cell function in responders at baseline and during treatment[40]. Previous findings also suggested that mregDCs may be associated with better patient outcomes, as confirmed by the presence of mregDC cell markers associated with improved overall survival in melanoma patients[72]. Furthermore, the enrichment of a CCR7 + DC signature (mregDC) has been linked to improved survival in lung cancer, cutaneous melanoma, breast, and colorectal cancer by analyzing 4,045 human solid tumor transcriptomes from the TCGA[73]. In our study, while we did not find mregDC proportion alone as a pan-treatment pro-survival biomarker, combining mregDC proportion and TCF7 + /− CD8 T ratio, a previously reported ICI response marker[9], allowed us to predict patient survival across treatments, suggesting a potential role for mregDCs in shaping and regulating T cell states, including TCF7 + CD8 + T cells, in human tumors in vivo. Notably, these two cell subtypes were recently found to be enriched in spatially organized stem-immunity hubs in human lung cancer, associated with response to immunotherapy[74].

Employing both scRNA-seq and snATAC-seq data from a substantial cohort of cDCs within the human melanoma TME, we meticulously delineated the three subtypes: cDC1, cDC2, and mregDC. Our investigation unveiled that mregDCs share transcriptional and epigenetic resemblances with cDC1s rather than cDC2s, suggesting their probable derivation from cDC1, in line with recent pan-cancer insights[75]. Alternatively, cDC2s may undergo more extensive epigenomic reprogramming during maturation, contributing to their plasticity in the TME. Recent studies have also indicated that a larger proportion of mregDCs may originate from cDC2, based on staining for CD141 (a cDC1 marker) and CD1c (a cDC2 marker)[45]. Determining the precise lineage of mregDCs in vivo will require lineage-tracing experiments. A key finding of our study is the identification of an mregDC-specific enhancer regulating IL15 expression. This enhancer, absent in cDC1/cDC2 subsets, provides a transcriptional blueprint for the elevated IL-15 production observed in mregDCs. Prior work established IL-15 as a survival signal for tumor-infiltrating CD8 + T cells[76], but our data now link this cytokine to a stable epigenetic program unique to mregDCs. Moreover, we pinpointed TFs with heightened activity in mregDCs, notably IRF1, IKZF1, KLF3, and KLF6, urging deeper exploration of their roles in orchestrating the mature and regulatory phenotype. Our analysis of transcriptional signatures linked with various cytokines revealed an enrichment of proinflammatory cytokines (IL1β, TNFα, IL12, IFNβ), alongside cytokines pivotal for DC maturation and activation (GM-CSF, TSLP, IL18) in mregDCs, aligning with expectations. Additionally, we noted an enrichment of leptin, LIF, and IL21 in cDC1 compared to mregDC, highlighting cytokines for future investigation. More importantly, we uncovered that

transcriptional alterations in tumor cells and fibroblasts correlated prominently with mregDC proportions in the TME, with both cell types upregulating ECM pathways in mregDC high samples. These findings gain significance as ECM proteins are known to influence DC maturation[77].

Overall, our study reveals a rich repository of single-cell profiles from human metastatic melanomas, coupled with extensive clinical data, and underscores the critical role of mregDCs in shaping the TME and their potential as prognostic markers and therapeutic targets in melanoma treatment.

## Limitations of study
In this study, our samples displayed significant clinical heterogeneity, including variations in treatment types and stages. While we attempted to address these variations computationally, a larger single-cell cohort would enhance the robustness of molecular and cellular correlates with patient responses and survival benefits. For communication analysis, integrating orthogonal data types like imaging and experimental approaches such as in vitro co-culturing systems, spatial transcriptomics, and in vivo tracer methods, would be invaluable in substantiating the predicted cell crosstalk.

## Methods
### Patient cohorts for single-cell analysis
The metastatic melanoma samples for this study were obtained from Mass General Brigham by G.M.B. under the protocol 11–181. This protocol was approved by the Dana-Farber Cancer Institute Institutional Review Board (PI: G.M.B.). Our study focused on annotated metastatic tumors treated with ICIs, targeted inhibitors, other immune therapies, or their combinations as primary treatment. Patient age, progression-free survival (PFS), and overall survival (OS) were calculated from the treatment start date. Lesional response was assessed using RECIST 1.1 criteria. No statistical method was used to predetermine sample size. Data from patients involved in unpublished clinical trials at the time of sample collection were excluded from the analyses.

### RNA/DNA extraction
We enzymatically digested and dissociated fresh tumor samples according to tissue dissociation kit protocols using the gentleMACS™ Dissociator (Miltenyi). Dissociated tissues were filtered, centrifuged, and the isolated cells were resuspended in 0.1% BSA in PBS and immediately processed for the generation of single-cell RNA (scRNA)/single-nucleus ATAC (snATAC) libraries using the droplet-based RNA/DNA sequencing technology.

### Single-cell RNA-sequencing and preprocessing
Briefly, 5000-6000 cells were profiled per sample using the Chromium Single Cell 3′ RNA reagent kit v3 (three samples were processed using kit v2) according to the 10X Genomics protocol. The generated cDNA libraries were indexed, pooled, and sequenced in batches using the NovaSeq 6000 S2 system and reagent kits (100 cycles; Illumina). We received the sequences, mapped the reads against the GRCh38 human reference genome, and quantified the read counts using the Cell Ranger pipeline version 3.1.0 (10x Genomics).

## Single-nucleus ATAC-sequencing and preprocessing

We performed snATAC-seq using the Chromium single-cell ATAC v1 chemistry from 10x Genomics with the target of 5000 nuclei per sample. Then the libraries were profiled using the Novaseq S2 (Illumina) technology. We aligned and quantified the libraries using the 10X cellranger-atac-2.0.0 pipeline.

## Single-cell RNA-seq data analysis

Quality control, dimensionality reduction, clustering and cell type annotation: We used CellBender[78] to remove ambient RNA in each sequencing library and called multiplets using Scrublet[79]. We then used the standard pre-processing workflow from Seurat (v4)[80] to prepare the scRNA-seq dataset. Single cells with <200 genes or >6000 genes or >10% of reads mapping to the mitochondrial genome were removed. Data normalization and scaling was performed using SCTransform[63], to adjust for sequencing depth and remove mitochondrial mapping percentage as a confounding variable. We then merged the libraries and performed principal component analysis (PCA) using the genes detected in all samples. After obtaining the low-dimensional embeddings, we performed Uniform Manifold Approximation and Projection (UMAP) on the top 40 principal components to visualize the dataset. We then constructed a k Nearest Neighbor (kNN) Graph in the top 40 principal components space and clustered cells using the Louvain algorithm. In all the datasets, we used the default parameter k = 20 and a resolution parameter ranging from 0.2 to 1.4 and proceeded with the lowest resolution that sufficiently highlighted the biological differences between clusters. We annotated each cluster using known cell-type-specific markers. To identify proliferating cells, we used the CellCycleScoring function to calculate the S phase score and G2M phase score for each cell. To identify tumor cells, we used inferCNV (https://github.com/broadinstitute/inferCNV) to infer copy number alterations from potential tumor clusters in comparison to normal cell clusters.

Cell type-specific subclustering: A cell type of interest was subsetted from the object containing all cells from the 36 samples, which passed quality control metrics described in the previous method section, using the subset function. We then ran SCTransform[63] on the raw counts of the subset, with the default "v2 regularization" and the "vars_to_regress" parameter equal to the percentage of mitochondrial genes. After applying SCTransform, PCA was applied on the scaled data. Next, to ensure integration of cells by cell subtypes rather than by the technical variation across samples, we used Harmony[81], an algorithm that projects cells into a shared embedding in which cells group by cell states rather than dataset-specific conditions, different samples in our case. Finally, non-linear dimensionality reduction was applied with UMAP, using the first 30 dimensions from the reduction from Harmony, to visualize the cells in 2D.

Cell subtype annotation: The identification of subtypes for each cell type was determined by two approaches: 1) a priori knowledge of specific markers for subtypes revealed in the literature, and 2) graph and/or density-based clustering algorithms to cluster groups of cells of the same cell type followed by identification of differentiating markers among clusters. Both approaches were used concurrently to find the best partitions of subtypes, as detailed below: 1) Using known markers for specific subtypes: Through extensive literature search, markers distinguishing cell subtypes were visualized using the Seurat function FeaturePlot to identify patterns of the expression of certain markers for the differentiation and assignment of subtypes. 2) Applying clustering methods to delineate subtypes: Clustering methods were applied to determine subtypes within cell type subsets and could be of either of the following methods: a) The FindClusters function in Seurat identified clusters of cells by constructing a shared nearest neighbor (SNN) graph from k-nearest neighbors (kNN). Then the modularity function was optimized using the Louvain algorithm, which finds non-overlapping clusters from networks to determine the clusters. The

default parameters were kept while varying the resolution to match the expected number of subtypes within the cell type (higher resolution = more clusters). b) Density-Based Spatial Clustering of Applications with Noise (DBScan), is a density-based clustering algorithm that uses an embedding of points (in our case the UMAP projection) to cluster points by how close or far they are from one another. The R package dbscan was used to find clusters, in which the "eps" parameter was set to the radius of neighboring points and the "MinPts" parameter, which specified the threshold for the minimum number of neighbors within the radius. By assigning points by these parameters, clusters as well as outlier points were found by the algorithm. 3) Validation of subtypes through differential gene expression analysis: Clusters could be compared against each other by finding DEGs between one or more clusters. We used FindMarkers to identify DEGs among the various clusters to verify that the clusters differentially express the known markers of interest from the literature or discover markers to confirm the presence of a subtype. This was applied when the marker expression was not very apparent merely by observation through FeaturePlot. Once clusters were determined, the subtype annotations were integrated back into the original object containing all cells from the 36 samples. Cells that passed quality control metrics but did not cluster with any annotated subtypes were found using dbscan. Outlier cells were discarded from the finalized object.

Subtyping of malignant cells: We used a different approach, as described in Gavish et al.[22], to uncover "meta-programs" of tumor cells, with each uniquely describing a distinct cellular state through a gene set. Each cell was assigned to the meta-program for which its sum of gene activities for the specific meta-program's genes was the maximum among all meta-programs tested and meeting defined constraints. Overall, a minimum score of 0.03 was enforced to yield 60.1% of tumor cells that would be assigned to a meta-program (to roughly match the threshold of 56.4% used by Gavish et al.[22]). After assignment, cells that did not reach the threshold for any of the meta-programs (default score for assignment = 1) or did not have the minimum number of genes to test for any meta-program (default number of genes = 25) were assigned as NA. The code to run the meta-program distribution is at https://github.com/tiroshlab/3ca/blob/main/ITH_hallmarks/MPs_distribution/MP_distribution.R.

Cell subtype proportion correlation analysis: Cell type and subtype abundances were calculated by sample. Relative subtype proportion was calculated by dividing the abundances of a subtype by its corresponding cell type. Given an input proportion by patient sample matrix, Pearson correlation was calculated for every pair of subtype proportions across samples using the Python corr function from the pandas library. Correlations of proportions where the cell type was not present in a sample were excluded from the analysis, as it would lead to a "division by 0" error. Groups of similar correlations were clustered together for the generation of the correlation matrices by measuring the distances between correlations and making clusters based on the distances, which was executed using the linkage and fcluster functions in the SciPy package in a wrapper developed by Yegelwel (https://wil.yegelwel.com/cluster-correlation-matrix/). After visual inspection of the correlation matrices, correlation plots highlighting the relationship between pairs of highly correlated subtypes were generated with an associated p-value to determine the probability of the result, assuming that the correlation coefficient was truly 0.

Cell subtype proportion by response analysis: To evaluate subtype proportions stratified by responding and non-responding tumors, we generated boxplots to visualize the distributions and measured the difference in distributions using the Wilcoxon rank sum test. This test was computed by the wrapper function wilcox.test in R and using a two-sided alternative (no prior information determining which distribution would be higher) to determine whether the distribution of a certain subtype proportion was different between

samples in the response or non-response groups. We also performed multiple analyses evaluating the distribution of subtype proportions on the different treatment group cohorts of samples from all treatments, samples treated with ICI, ICI-only treated samples, and samples treated with anti-PD1. *P*-values were labeled if it was less than 0.1, but we still evaluated significance based on a *p*-value less than 0.05.

Kaplan Meier survival analysis: For our analysis, survival curves between two groups of samples were compared, and their differences were evaluated for significance using the log-rank sum test. Kaplan Meier survival curves were generated in R using the libraries survival and ggsurvfit. The function Surv from survival combined the survival data and corresponding censored data, which was used as the response variable of the formula survfit2 model from ggsurvfit. The survival data was stratified by the proportions we evaluated survival by, which was passed as binarized values according to the chosen cutoff value with which to group samples. Log-rank testing was performed for each survival analysis, and *p*-values were calculated between the different groups using the internal function add_pvalue in ggsurvfit.

DEG and pathway analysis of cDC subtypes: To perform differential gene expression analysis comparing mregDC to the other cDC subtypes in a pairwise fashion, we applied FindMarkers on the SCTransform normalized gene expression of the cDC scRNA-seq subset and compared the subsets of mregDC and cDC1 or mregDC and cDC2 using a Wilcoxon rank sum test. A log fold change threshold of 0.01 was used, and the minimum percentage of cells with the detected gene was set to 10%. DEGs were selected using an adjusted *p*-value (Bonferroni method by default) of 0.001. We used Metascape[82] to identify the enrichment of pathway networks for the DEGs.

Trajectory, regulatory network and immune response inference for cDC subtypes: We leveraged the CytoTraceKernel function in the Python package Cellrank to infer the CytoTrace pseudotime[83]. We analyzed the activated regulatory networks in cCD1, cDC2, and mregDC subtypes with SCENIC[48]. Starting with the raw count matrix as input, we used GRNBoost2 to find regulatory networks, which are sets of genes co-expressed with transcription factors (TFs). To remove false positives and indirect targets, we pruned the networks using RcisTarget, which identifies direct-binding targets with motif enrichment of the correspondent upstream regulator. Finally, we used AUCell to compute the regulon activity score (RAS) in each cell based on the ranked expression value of genes in the regulon. To select regulons specific to mregDCs, we calculated the regulon specificity score, which reflects the entropy of RAS in cells across cDC subpopulations, for each detected regulon. The regulon specificity score ranges from 0 to 1, with a higher value indicating more specificity of the regulon. We then used the Immune Response Enrichment Analysis (IREA) tool to find enriched cytokine expression in mregDCs relative to cDC1s and cDC2s. We used DEGs between mregDCs relative to cDC1s and cDC2s as defined above (Supplementary Data 6). Then, the upregulated and downregulated DEGs were separately input into IREA, with "mregDC", "cDC1" or "cDC2" selected as the cell type and score chosen as the method. We combined the outputs from both sets of DEGs, negating the enrichment score of the downregulated DEGs, and then filtering for significance (adjusted *p*-value <=0.05). We removed cytokines that were determined to be significantly expressed in both the upregulated and downregulated DEGs.

Cell-cell communication for cDC subtypes: To infer cell-cell communication, we used CellPhoneDB[57], which takes in single-cell transcriptomic data and compares gene expression values to a database of ligand-receptor interactions to statistically test the significance of those interactions. We performed CellPhoneDB analysis to find ligand-receptor pairs between the cDC subtypes (mregDC, cDC1 and cDC2) and all other subtypes existing in our dataset. To preserve the integrity of true interactions, CellPhoneDB was run by sample (35 times independently for each sample in our scRNA-seq dataset). A log-normalized RNA count matrix of the sample and the list of subtype

annotations for each cell were given as inputs to the program. Using the statistical analysis method of CellPhoneDB (Method 2), interactions between subtypes were obtained and compared to a null distribution generated by random shuffling of the cell types into clusters. The method is sped up via a geometric sketching procedure[84]. All parameters were set to the default. We used the output of the significant_means field, which provided an average mean value for an interaction between two partners of two cell subtypes if the interaction was significant; otherwise, the mean was equal to 0. We then looked into the interactions between the cDC subtypes and the other cell types. With our interest being in mregDCs, we had two goals: 1) to find interactions that existed only between the mregDC subtype and non-cDC cell types and not between cDC1 or cDC2 and non-cDC cell types; and 2) find cDC to non-cDC interactions that were differentially expressed between the cDC subsets. Several thresholds were applied for each analysis: 1) Exclusive interactions were only considered if the mregDC to other non-cDC subtype interaction was significant in at least 3 samples. 2) To evaluate differentially expressed interactions, the Wilcoxon rank sum test was applied for the same ligand-receptor interactions expressed by cDC1, cDC2, and mregDCs. Wilcoxon rank sum *p*-value < 0.05 and the interaction being present in more than half of the samples in each group (>5 samples for mregDC high, >13 samples for mregDC low) were used to call the differential. This analysis was performed separately when evaluating interactions when the ligand was expressed on cDCs (cDCs as the source) or when the receptor was expressed on cDCs (cDCs as the target).

## Single-nucleus ATAC-seq data analysis

Quality control, clustering and dimensionality reduction: For quality control, we kept cells with peak region fragments between 5000 to 20,000 nucleotides, transcription start site enrichment score > 4, and percent reads in peaks > 20%. We quantified the activity of each gene in the genome using the GeneActivity function in Signac[85], which sums the fragments in the gene body coordinates, including 2 kb upstream region, according to Ensembl annotation EnsDb.Hsapiens.v86. For dimensionality reduction, we performed term frequency inverse document frequency (TF-IDF) normalization and singular value decomposition (SVD) on the peak-by-cell matrix. To visualize the dataset in 2-dimensions, we performed UMAP using the top 50 components and used graph-based clustering to find clusters. With a combination of known markers and label transfer from scRNA-seq data, we assigned cell labels to each snATAC-sequenced cell.

Transfer of scRNA-seq annotations: Given our fully annotated scRNA-seq object of cell types and subtypes of the 36 samples derived from the methods described above, we found anchors between the RNA object and the object of the 14 matching snATAC-seq samples. Label transfer was first performed at the cell-type level, and performed a second time to label subtypes within each cell type-specific subset. First, gene activity scores derived above were compared to scRNA-seq gene expression counts via canonical correlation analysis (CCA) using the FindTransferAnchors function with the reduction parameter set to "cca" and the first 2000 variable features used from the scRNA-seq dataset. The resulting anchors found were used to predict cell type (or subtype) labels for each cell in the snATAC-seq dataset using the TransferData function, with the reference data parameter set to the scRNA-seq cell types or subtypes and weight reduction set to the latent semantic indexing (LSI) projection of the snATAC-seq data. Predicted labels were assigned to each cell and visualized on the UMAP of the 14 snATAC-seq samples. For subtype label transfer, we repeated the data normalization and dimensionality reduction steps mentioned above for each subsetted cell type.

RNA and ATAC co-embedding for cDCs: After subsetting cDCs from both snATAC-seq and scRNA-seq datasets, we used FindIntegrationAnchors in Signac[85] to identify co-varying components of the snATAC-seq and scRNA-seq datasets. Then we leveraged CCA to co-

embed cells from both the transcriptomic and epigenomic space using the identified canonical 'basis' vectors. The co-embedding in shared lower-dimensional space maximizes correlation between the two datasets, thus allowing joint analysis and cell-typing of scRNA-seq and snATAC-seq datasets. We re-assigned the snATAC-seq cells with cDC subpopulation labels from their mutual nearest scRNA-seq neighbors in the shared space, compared with the cell subtype label transfer results derived above, and decided on the final subtype labeling of snATAC cells based on consensus and marker gene activities.

Differential accessible region analysis for cDCs: DARs were identified in a similar fashion to DEGs. Using normalized values of the peaks of cDCs in our snATAC-seq data, DARs were found using FindMarkers, again comparing the identity classes of mregDC and cDC1 or mregDC and cDC2. Logistic regression was used to identify DARs, as it corrects for the sequencing depth by incorporating the total number of fragments as a latent variable. A log fold change threshold of 0.01 was used, with a minimum percentage of the same peaks detected in both classes set to 5% to increase sensitivity relative to DEG analysis because of the sparse nature of ATAC data. DARs were labeled with the closest gene to the region using the ClosestFeatures function in Seurat. DARs were selected using an unadjusted $p$-value of 0.01 due to a lack of signal using adjusted $p$-values. Thus, interpretations of DARs are not conclusive and merit further investigation and validation.

Global correlation, pathway and motif enrichment analyses for cDCs: To compute cell type similarities to migDCs, we first calculated the average peak accessibility for each cell type of each peak accessible in migDCs. Then, we calculated the Pearson correlation of the average peak accessibility between the cell type and migDCs. To investigate cis-regulatory regions, we used the Genomic Regions Enrichment of Annotations Tool (GREAT)[86]. We input significantly upregulated and downregulated regions independently into GREAT, yielding GO biological processes. For processes identified to be enriched in both upregulated and downregulated regions of the same comparison, the processes were removed. Significant GO biological processes were visualized using clusterProfiler[87] and enrichplot[88]. To investigate differentially accessible motifs between cDCs, we added motif information using JASPAR2022 and BSgenome.Hsapiens.UCSC.hg38. Then, we computed the per-cell motif activity score using Signac's implementation of chromVAR. We used FindMarkers in the chromVAR assay to find the resulting differentially accessible motifs.

## Integration with previous single-cell studies

Tirosh et al.[7], Jerby-Arnon et al.[8] and Sade-Feldman et al.[9] had gene count matrices given in transcripts per million (TPM). Zhang et al.[10] and Biermann et al.[11] provided raw count matrices. We assumed that merging log-normalized TPM and raw count matrices would capture the differences in relative expression, and the variation of gene expression would be preserved, which was proven to be true. As each count matrix in their available data format from each study was converted into a Seurat object, we performed integration feature selection with SelectIntegrationFeatures and anchor discovery using log normalization and canonical correlation analysis (CCA) reduction with FindIntegrationAnchors. After obtaining the set of anchors, we integrated the six objects using IntegrateData. For label transfer, another set of anchors was found from our study's object as the reference and applied to a subset of the integrated object corresponding to the five studies as the query using PCA reduction of the reference. Predicted cell types and subtypes of the scRNA-seq of the five studies were predicted in a similar fashion to the label transfer of cell types and subtypes from scRNA-seq to snATAC-seq as described above. Subtypes were found after the annotation of cell types. For the construction of the final UMAP of the integrated object of all studies, the default assay was set to "integrated" and then scaled using the function ScaleData. PCA was applied using the variable features of the integrated object, and RunUMAP was executed on the first 30 PCs of the PCA reduction. No further batch correction was needed as

cells from different studies were well integrated and cell types were clearly separated on the UMAP embedding.

## Identification of cell states and DEGs associated with mregDC proportions

**Algorithm.** When performing DEG analysis between samples, we aim to explore which cell state is associated with the mregDC groups and what DEGs characterize this cell state. Existing methods for detecting DEGs, such as pseudo-bulk and mixed models, assume homogeneous populations when comparing subtypes and treating all cells from one sample equally. Hence, the analysis still operates at a sample level and does not leverage the between-cell heterogeneity available in the scRNA-seq data. Our assumption is that, instead of the whole cell population in a group, the mregDC proportion affects certain binary cell states in a cell type, e.g., exhaustion, activation, apoptosis, etc, and we aim to extract the mregDC-associated cell states and identify the DEGs in these cell states. To address this, we performed the DEG analysis in two steps: 1) identify cell states associated with the mregDC groups, 2) identify DEGs for this cell state. In Step 1, we developed a factorization algorithm to find the binary variable that explains the most variability of the count matrix with the adjustment for the covariates and sample-level heterogeneity under a Negative Binomial mixed model as in NEBULA[89]. Conceptually similar to the top principle component, this identified binary variable is the major cell state of the cell type. Our algorithm also returned the uncertainty of the estimation of the cell state and this uncertainty was used in the following steps to conduct valid testing for DEGs. In Step 2, we then performed an association test between the extracted cell state and the mregDC groups in each cell type or subtype to identify the mregDC-associated state using a logistic mixed model. For those mregDC-associated states, we obtained their DEGs using the Negative Binomial mixed model implemented in NEBULA[89] and further took into account the uncertainty in the cell state estimation. We predict that this algorithm would find the cell states that drove the difference observed between the mregDC high group and the mregDC low group if mregDC is the major underlying factor driving the differential expression, and the DEGs would be the potential regulators of the mregDC phenotype, especially at the cell-type level.

**Model building.** The model takes in two pieces of data: first, the raw gene-by-count matrix of the samples, and second, the metadata associated with each sample, which is used as the design matrix. First, we compiled the raw count matrices of all samples, which were subsetted by cell type or cell subtype. Then we created the design matrix, which included the mregDC group variable, which was 0 for an mregDC low sample (mregDC/cDC ≤ 0.185) or 1 for an mregDC high sample (mregDC/cDC > 0.185). The design matrix also included the sample-level variables to account for. We selected the sample-level variables of age, sex, tissue of biopsy (skin, lymph, other), treatment state (pre, on, post), and treatment group (other, with ICI, ICI only, anti-PD1) and the cell-level variables of mitochondrial gene percentage and ribosomal gene percentage. Sample or cell-level variables for which all cells in the subset had the same value were removed from the design matrix prior to running NEBULA. The offset was set to the number of counts of RNA, i.e., the total number of molecules detected within a cell. NEBULA[89] was first run for each cell type and subtype to estimate the overdispersions, in which the count matrix, offset term, and design matrix were inputs to the model. Then, given the overdispersions, a factorization algorithm for the top binary factor was run for each cell type. We used the factors called from this step as the major cell states for cell subtypes and tested its association with the mregDC group. For mregDC-associated cell states, we used NEBULA[89] again to obtain the DEGs associated with these states.

Pathway analysis for identified DEGs: We used the fGSEA (fast gene set enrichment analysis) package in R. We narrowed down the

gene set of canonical pathways to test enrichment, which was downloaded from https://www.gsea-msigdb.org/gsea/msigdb/human/collections.jsp. The list of all genes (significant and not significant) with their associated log fold change values were used as input to fgsea, setting the pathways to the downloaded canonical pathway gene set and testing for positively and negatively enriched pathways (scoreType parameter = 'std' for standard). Given the output of pathways, we identified main pathways, which are pathways that are independent from each other, using the function collapsePathways and an adjusted p-value less than 0.05 using the Benjamini-Hochberg correction, which was the default. The top pathways with the highest enrichment scores and adjusted p-value < 0.05 were visualized using aPEAR (Advanced Pathway Enrichment Analysis Representation)[90], a separate package in R that can generate enrichment networks from pathways that are similar to each other. All the canonical pathways generated from the cell type or subtype subsets were used as input, together with their adjusted p-values, normalized enrichment score, the number of genes in the pathway, and the list of genes in the pathway. Using a similarity metric to cluster redundant pathways, aPEAR assigns a general name for the clusters of pathways using the PageRank algorithm[90]. All default parameters were used in the clustering and visualization of the pathways.

### Bulk RNA-seq meta-cohort data analysis

To determine whether or not a signature of mregDCs is able to stratify an independent cohort of 318 ICI-treated patients with metastatic melanoma and with RNA-seq data[42], we used ssGSEA to score the mregDC signature (886 genes positively define mregDCs, Supplementary Data 3) in the bulk RNA-seq data set, and then performed response comparisons and survival analysis the same as described in the scRNA-seq section.

### Analysis of prior spatial transcriptomics data

To quantify the ECM-mregDC co-localization on slide-seqv2 sample ECM06 published by Biermann et al.[11], we computed the co-occurrence score between myeloids and fibroblasts (per published annotation) using the squidpy.gr.co_occurrence() function. We scored the mregDC signature (Supplementary Data 3) in beads labeled as 'myeloid cell', and considered the upper 20% scored beads as 'mregDC-high' and the lower 20% scored beads as 'mregDC-low'. Likewise, we scored the reactome NABA-matrisome pathway in beads labeled as 'fibroblast cell', and considered the upper 20% scored beads as 'ECM-high' and the lower 20% scored beads as 'ECM-low'. We labeled all other beads in the slide as 'other'. The co-occurrence analysis showed the probability of observing 'ECM-high' or 'ECM-low' fibroblasts within a 1000 radius when conditioned on the presence of a 'mregDC-high' cell.

### Immunofluorescence

Human sentinel lymph node FFPE block was obtained from Rutgers Cancer Institute Biospecimen Repository and Histopathology Service Shared Resource Center at Rutgers University. Five-micrometer thick sections were cut using a microtome and transferred onto Superfrost Plus slides (Thermo Fisher) and allowed to dry for 10 min at room temperature, followed by 30 min in an incubator at 65 °C. The sections were dewaxed, dehydrated and high-temperature citrate buffer antigen retrieval performed, then blocked with goat serum for 30 min at room temperature. Sections were incubated with rabbit anti-human CCR7 (1:200, Proteintech), Armenian hamster anti-human CD11c (1:100, Novus Bio, clone AP-MAB0806), mouse anti-human IRF4 (1:300, Thermo, clone 3B1D2) and polyclonal goat anti-human IRF8 (1:200, Novus Bio) antibodies overnight at 4 °C followed by goat anti-rabbit AF750 (1:2000, Thermo), goat anti-armenian hamster AF488 (1:1000, Thermo), goat anti-mouse IgG2a AF594 (1:2000, Thermo) and donkey anti-goat AF555 (1ug/ml, Thermo) for 1 h at room temperature, with PBS washing between incubations. Appropriate single-stained controls for each antibody were run in parallel. After staining, slides were mounted with Prolong Glass anti-fade with Nuc-Blue stain (Hoechst 33342) (Thermo), cured for 24 h at room temperature in the dark and imaged using Leica Stellaris8 confocal microscope using 40x objective and the dedicated LAS X software at Waksman Institute Shared Imaging Facility, Rutgers, The State University of New Jersey.

### Reporting summary

Further information on research design is available in the Nature Portfolio Reporting Summary linked to this article.

## Data availability

The raw and pre-processed scRNA-seq data generated in this study have been deposited in the GEO database under accession code GSE269936. The raw and pre-processed snATAC-seq data for cDCs generated in this study have been deposited in the GEO database under accession code GSE303948. The processed scRNA-seq and snATAC-seq data are deposited in the Zenodo database under accession code 15603513. The scRNA-seq publicly available data used in this study are available in the GEO database under accession code GSE72056[7], GSE115978[8], GSE120575[9], GSE215119[10], and GSE200278[11]. The meta-RNA-seq publicly available data used in this study are available in supplementary tables of the corresponding study or as.rds objects available on GitHub [https://github.com/davidliu-lab/diffIO-hypoxia-figures][42]. The remaining data are available within the Article, Supplementary Information, and Source data.

## Code availability

All codes that are necessary to reproduce all the results in the paper are implemented in Python and R and are publicly available at GitHub (https://github.com/KellisLab/scCancer).

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

## Acknowledgments

We thank past and present members of the Boland, Kellis, Liu, and Rai laboratories for thoughtful scientific discussions. We thank Natalie D'Amore, Eric Lightcap and Sarah Hesse for discussions and feedback on sample collection, sample processing and data analysis. This work was supported by Takeda Pharmaceuticals, Boston, MA, USA (M.K.), the Adelson Medical Research Fund (G.M.B.), Bill and Emma Roberts MGH Research Scholar (G.M.B.), the Society for Immunotherapy of Cancers (D.L.), the Doris Duke Charitable Foundation Clinical Scientist Training Program (D.L.), and Rutgers, The State University of New Jersey (J.Y.).

## Author contributions

This study was designed and directed by J.Y., G.M.B and M.K. D.T.F., T.S., and R.J.S. compiled clinical information. L.H., K.G., D.T.F., and S.W. coordinated sample acquisition. L.H. and K.G. performed scRNA-seq and snATAC-seq experiments. J.Y., C.W., D.F., L.C., J.G., J.F., A.Y.H., E.J.R., and R.T. performed data processing and computational analysis. L.H. designed the algorithm used in the association analysis with mregDC proportions. A.Y.H. performed analysis in the bulk meta-cohort. M.A. performed the immunofluorescence experiment. D.L. designed and directed the bulk meta-cohort study. S.K.Y., W.D., K.P.B., and K.R. provided scientific feedback. J.Y., C.W., D.F., G.M.B., and M.K. wrote the manuscript.

## Competing interests

The authors declare that they have no conflict of interest.
