## [Transparent Peer Review file · Nature Communications]

Mature and migratory dendritic cells promote immune infiltration and response to anti-PD-1 checkpoint blockade in metastatic melanoma

Corresponding Author: Dr Jiekun Yang

Version 1:

Reviewer comments:

Reviewer #1

(Remarks to the Author)

Using scRNA-seq and snATAC-seq data, the authors have characterized the immune, non-immune, and tumor cell diversity of metastatic melanoma from patients, including 22 patients treated with immune checkpoint inhibitors (ICIs) and 3 patients not treated with ICIs. The main finding of the study is the identification that mature dendritic cells enriched in immunoregulatory molecules (mregDCs) are enriched among conventional dendritic cells (cDCs) upon anti-PD-1 treatment response. A high proportion of mregDCs also correlates with better progression-free survival (PFS) in patients receiving combination therapy plus anti-PD-1, and the mregDC signature in ICI-treated bulk samples correlates with better PFS. The epigenomic landscape of the cDC to mregDC transition is also nicely described using snATAC-seq. Overall, the study is well done and helps to advance our understanding of specific components (cell types, cell states) that correlate with ICI response. The data from this ICI cohort will be of particular interest to the community. The demonstration that mregDC proportion or signature serves as a good prognostic factor for progression-free survival in ICI-treated melanoma could advance tailored immunotherapy interventions. However, there are still some issues that need to be addressed to better substantiate the claims. The authors should carefully address all comments.

Major Comments

1. Analysis of Pre-, On-, and Post-Treatment Biopsies. Although the authors have used pre-, on-, and post-treatment biopsy samples, they have not conducted comparative analyses between these time points. The authors should emphasize the immune population shifts resulting from the treatment by comparing these samples.

2. Definition of Immune Cell Diversity (Figure 2).

- Myeloid and T Cells (Figures 2C, D, E): The clusters should be annotated with marker genes, including their average expression levels and possibly the number of unique molecular identifiers (UMIs) per cell. This information would be particularly useful for distinguishing macrophage clusters (see comment below) and differentiating between natural killer T (NKT) cells and certain T cell subtypes that are challenging to separate.

- Macrophage Compartment: The macrophage compartment cannot be transcriptionally separated into M1/M2 subtypes, as this is an in vitro dichotomy. The authors should refine their clustering using up-to-date definitions, such as resident-like macrophages, TREM2+ macrophages, and different subtypes of tumor-associated macrophages (TAMs) as recently described in pancreatic cancer (Dunsmore et al.). The lack of proper definitions could influence the study's findings. Similarly, the diversity of B cell populations should be increased if possible.

- Function Inference from Gene Expression: Anti-tumor or pro-tumor functions cannot be inferred solely from gene expression within a cluster or from the enrichment of a cluster itself. Such assertions should be removed throughout the paper. Only correlations with response, overall survival (OS), or progression-free survival (PFS) should be presented.

- Correlation of Cell Subtype Proportions: Correlations of cell subtype proportions should be performed on immune cells only. Introducing tumor cell subtypes can drive the correlation and potentially mask the immune archetypes that could be

observed. Figure 2F shows that the correlations of cell subtypes are completely different in another cohort; performing immune-only correlations of cell subtype proportions could help clarify this. If the correlations remain different, the authors should discuss these discrepancies.

3. Association of mregDC with Response (Figure 3):

- For all the clusters identified in Figures 1 and 2, the authors should present in a single figure which clusters are associated with response, which are associated with non-response, or which are not significant.

- Regarding the observed enrichment of plasma cells in non-responders instead of responders, the authors should discuss the accumulating evidence linking plasma cells and response to checkpoint blockade, such as in the study by Helmink et al. (Nature, 2019; <https://www.nature.com/articles/s41586-019-1922-8>).

- Kaplan-Meier Analysis (Figures 3B and 3C). Please ensure that the Kaplan-Meier graphs are computed per patient rather than per sample.

- mregDC Gene Signature (Figures 3D and 3E): It is unclear what mregDC gene signature was used, even after examining Supplementary Table 3, "Non-tumor cell subtype markers in reference to all the other cells." We do not know the cutoff in terms of the number of marker genes used to define the mregDC signature. This information is crucial if this signature is to be used in future predictions of response to immune checkpoint blockade. The signature and the number of genes used should be clearly presented in the text, figures, or tables.

4. Transcriptional Landscape of cDCs

- The authors assert that mregDCs express higher levels of IRF4 compared to cDC2. This should be validated at the protein level. This can be achieved by using cryopreserved cells from the cohort and performing intracellular staining with an anti-IRF4 antibody, comparing DC1 CD141+ CCR7+ (mregDC) and – (non mregDC) and DC2 CD1c+ CCR7+ (mregDC) and – (non mregDC).

- It cannot be concluded from the current data that most melanoma mregDCs are derived from DC1. Demonstrating this would require fate mapping experiments in mice or lineage tracing in patient-derived organoids (which we are not requested for this revision). The authors should modify this claim or discuss. Previous CITE-seq experiments in non-small cell lung cancer and hepatocellular carcinoma (Maier et al. Magen et al.) concluded mregDCs originate from both DC1 and DC2, but that likely more DC2 based on CD1c versus CD141.

5. Epigenomic Landscape (Figure 5)

- Given the 43,499 single nuclei obtained, they should provide an epigenomic atlas of the other populations, not just the cDCs. They can show epigenetic changes within T cells (e.g., during exhaustion) and other myeloid cells, such as monocytes and macrophages. It would also help to define the changes in accessible chromatin regions between responders and non-responders. Furthermore, presenting the epigenomic landscape across the populations co-enriched with mregDCs in responders versus non-responders would make for a better Figure 6 than the current one, which presents findings that are difficult to leverage in terms of basic or clinical significance.

Minor Comments

- The introduction includes a list of previously published scRNA-seq studies, which is unnecessary given that Supplementary Figure 1D already provides a comparison. Instead, the authors should focus on providing conceptual points that introduce the major findings of the study, such as details on the dendritic cell epigenome or background information on TCF1+ CD8+ T cells.

- Clarification on cDCs in Immune Surveillance: The statement, "While immature cDCs are essential for immune surveillance, their maturation is vital for immunogenicity involving antigen processing, presentation, and T-cell co-stimulation," requires correction. The role of cDCs in immunosurveillance is predominantly attributable to mature cDCs.

- Relevant Citation: When mentioning that "Although prior studies associated a higher density of mregDC-like dendritic cells with improved survival in cutaneous melanoma, this state remains poorly characterized in melanoma across different treatments," the authors should cite: Barry KC, et al (PMID: 29942093), which demonstrates the positive prognosis of CCR7+ DC1 (mregDCs originating from DC1) in melanoma during PD-1 immunotherapy.

- The table in Figure 1A should be simplified to present data per patient rather than per sample to enhance clarity.

- In figure 1B The treatment group, treatment state, and lesional response information should be moved to the supplementary material.

- Given that the study primarily focuses on immune clusters, Figures 2A and 2B should be placed in the supplementary section.

- In Figure 3 and Associated Extended Figures the proportions of mregDCs and cDCs should also be depicted among all MNPs to provide a comprehensive view of their distribution.

- The statement, "mregDCs [...] play an important role in mediating ICI effects in mouse models²⁷," should be rephrased. The referenced study primarily demonstrates the modulation of PD-L1-expressing CD11c⁺ cells. While it is plausible that mregDCs play a role in this context, this has not been formally proven in that particular paper.

- It should also be noted that both human and mouse mregDCs express CCL22 and CCL17, as reported in the literature—not just mouse mregDCs.

Reviewer #2

(Remarks to the Author)

In this manuscript, the authors aim to delineate the complex landscape of the tumor microenvironment (TME) in metastatic melanomas. They performed single-cell RNA sequencing on approximately 189,000 cells across 36 samples and single-nucleus transposase-accessible chromatin sequencing on around 43,000 cells from 14 samples in a metastatic melanoma cohort. They identified 55 immune, tumor, and stromal subtypes, including a co-enriched population module consisting of mregDC and naive T and B cells. They found that mregDC abundance was associated with better progression-free survival (PFS) in ICI-treated patients and a potential synergistic effect with the TCF7⁺/CD8 T cell ratio in predicting patient survival. Subsequently, they performed cell type/subtype-specific gene, pathway, immune response enrichment, and ligand-receptor interaction analyses for mregDCs, suggesting their roles in enhancing the immune response against cancer.

Major Concerns:

While this dataset is valuable and the authors tried to address an important knowledge gap in the cancer immunity related to mregDCs, the analysis lacks depth. Major revisions are needed to integrate or connect different parts of their analysis, such as tumor co-expression module analysis, cell subtype co-enriched module analysis, DEG, interactome and pathway analysis, and snATAC analysis of mregDC vs cDC1/cDC2. Additionally, the survival analysis lacks necessary controls for many confounding factors, casting doubt on the robustness of their conclusions regarding mregDC as a prognostic predictor of patient survival.

Detailed Points:

1. In Figure 2, the authors assigned tumor cells to 15 pan-cancer meta-programs previously compiled. However, since this is a melanoma dataset, they should also consider conventional melanocytic vs. dedifferentiated subtypes, the latter being resistant to targeted therapy and immunotherapy (<https://doi.org/10.1016/j.ccell.2018.03.017>).
2. Cell Subclustering Analysis: How is this TAM distinct from M1 and M2 (Figure 2C)? It expresses classical M2 markers such as MRC1 and CD163. Additionally, markers mentioned in the manuscript, like VEGFA, IL-10, TNF, and TGFB1, were not shown in Extended Figure 2B. For B cells, why is the plasma cell called regulatory?
3. MALAT1 can indicate low-quality cells, so the authors should be cautious about their definition of HNRNPH1⁺ T_{eff} cells in Fig 2C (<https://kb.10xgenomics.com/hc/en-us/articles/360004729092-Why-do-I-see-high-levels-of-Malatl1-in-my-gene-expression-data>). The same check should be done on the high level of heat shock protein - is this real subset or artifact of single cell RNAseq?
4. Correlations of cell subtype proportions (Figure 2F): The annotation of the modules is dubious. For example, why is module 1 called "Anti-tumor immune infiltration"? This is misleading because the module represents groups of subtypes correlated in their relative proportion across samples; there is no immunofluorescence data showing that the immune subtypes in module 1 specifically infiltrate the tumor parenchyma. Module 6 seems to have more tumor-infiltrating features with multiple effector T cell subtypes and interferon response in the tumor population. Also, how is module 4 an anti-tumor module with multiple proliferating tumor cell populations?
5. The statement of "conserved cell subtype modules regardless of treatment" (page 4, Figure 2F, Supplementary Figure 2A) is questionable. Among the 36 samples, 30 were treated with ICI, so it is not surprising that the overall correlation modules pattern is dominated by ICI-treated samples. Indeed, Supp Fig 2C looks quite different (is this atlas analysis include Sade Feldman's?) The authors should perform separate module analyses for ICI-untreated and -treated samples to see if the correlation module pattern changes.
6. Association between mRegDC and ICI response. Did the authors compare the differences in monocyte, macrophage and DCs among R and NR to ICI against Sade Feldman Cell 2018 dataset?
7. Survival analysis in Figures 3, Extended Figure 3 and Supplementary Figure 3: This section raises several concerns. The authors compared the relative fraction of different cell subtypes between ICI responders (R) and non-responders (NR). However, among the 19 ICI-treated samples, 7 are pre-treatment, 4 are on-treatment, and 5 are post-treatment (are these ICI resistant?). Combining pre- and post-treatment samples for comparison is questionable because the treatment itself could have altered the tumor transcriptome profile and cell type composition.
8. For the survival analysis, beyond Kaplan-Meier plots, the authors should perform multivariate analysis to account for potential confounding clinical factors (e.g., treatment type, treatment state, days to last ICI treatment) to show whether mregDC is an independent prognostic predictor.

9. Additionally, there are discrepancies in sample numbers between different figures. Figure 3A, B shows data from 18 combo or anti-PD1 samples, but Figure 1A shows there are 19 combo or anti-PD1 samples; Figure 3C shows a "total" of 33 samples, but the cohort is 36 samples.

10. Synergy Between mregDC and TCF7⁺/CD8 T Ratio: One interesting finding is the potential synergy between mregDC and TCF7⁺/CD8 T ratio in predicting patient survival. I can only see "high" and "low" stratification. What is the definition of "high/low"? I would expect to see 4 strata here.

11. The authors should conduct more in-depth analysis of this angle. For example, do mregDC and TCF7⁺ CD8 T cells (which of the T cell population in Fig 4G is TCF7 high?) show heightened interactions of antigen presentation (cellchat), co-stimulatory pathway or other T cell/DC related pathways within the TME?

12. The validation of mregDC signature's prognostic power in an independent bulk RNAseq dataset (Figure 3D) is not convincing because the mregDC signature score in bulk RNAseq could reflect overall immune infiltration, which is associated with better survival and ICI response. The authors should provide more detailed analysis to show whether other immune subtypes or overall immune infiltration levels (using tools like ESTIMATE) are also associated with longer PFS/OS.

13. Comparison with Previous Findings of mregDC: As a data-analysis-based paper with no functional in vitro and in vivo validation, the authors should compare their findings with previous studies on cancer immunity and mregDC-T cell interactions (e.g., PMID: 38728412, 34343496, 37708889). For example, is there heightened CD40:CD40L stimulation of CD4 T cells by mregDC? How about antigen presentation or cross-presentation pathways? What about DC-derived chemokine signals (e.g., CXCL16, CXCL9, CXCL10, IL-12, IL-15)?

14. snATAC-seq Analysis: The authors identified enriched TF motifs of five NF-κB TF family members in mregDC. How do these results reconcile with the scRNAseq analysis? Among the DEGs identified in scRNAseq, do any show corresponding more accessible chromatin regions and correlation with TF motif enrichment in these regions?

Reviewer #3

(Remarks to the Author)

Reviewer #4

(Remarks to the Author)

Yang et al provides a comprehensive single-cell RNA and ATAC-seq atlas for metastatic melanoma tumor microenvironment across 36 samples in total. This is a great addition to the melanoma community where single-cell multiomics cohorts with immunotherapy responses are still missing.

The study explores unique cell subsets and their functional characteristics, and emphasizes the presence of mregDC as the predictor of anti-PD-1 blockade response. This finding is somewhat expected as studied extensively elsewhere (see review, <https://onlinelibrary.wiley.com/doi/10.1002/ctm2.1199> for example). The interesting finding is stromal correlations of mregDC, and suggesting the stromal influence on DC maturation. While there are increasing appreciations towards iCAF/mCAF paradigm to crucially influence the immune microenvironment, it seems that the current manuscript somewhat halted its deeper dive towards studying the spatial architectures around mregDC-rich tumor regions, and verify its findings on ECM pathway enrichments. Certainly, there are existing spatial transcriptomes from Biermann et al. 2022 study to at least confirm the ECM-mregDC hypothesis. It is imperative for this study to validate this hypothesis with additional spatial data.

Overall, the current manuscript entails an interesting data set, yet the convoluted exploration of diverse cell subsets substantially degrade the readership. While the take-home message seems to be the predictive power of mregDC presence for immunotherapy response, this still lacks the novelty as it stands now. It looks that the manuscript requires additional study to realize the full potentials of the finding such as validating the ECM-mregDC hypothesis on additional spatial data sets that are publicly available.

Some minor comments are:

- Some schools of bioinformaticians regards over-expressions of long non-coding RNAs such as MALAT1 as a cell quality metric (see <https://www.biorxiv.org/content/10.1101/2024.07.14.603469v1>). Although I agree these genes have functional implications in the immune subsets as discussed in the manuscript,
- Lack of Biermann et al. 2022 study ([https://www.cell.com/cell/pdf/S0092-8674\(22\)00712-7.pdf](https://www.cell.com/cell/pdf/S0092-8674(22)00712-7.pdf)). It provides comparative study of single-cell/-nuclei transcriptomes for brain and non-brain metastatic melanoma for 32 samples in total.
- If mregDC enhances immune infiltration, I believe it should coincide with the absence of T-cell exclusion program in non-responsive tumors in the tissue.
- multivariate survival analysis to analyze the predictive power of mregDC abundances along with other covariates (gender, metastatic sites, TCF⁺/CD8 T-cell ratios etc).
- Incorporating additional bulk tissue cohorts for validation (Liu et al. 2019: <https://www.nature.com/articles/s41591-019-0654-5>, Riaz et al. 2017: [https://www.cell.com/cell/fulltext/S0092-8674\(17\)31122-4?](https://www.cell.com/cell/fulltext/S0092-8674(17)31122-4?)

Reviewer #5

(Remarks to the Author)

The manuscript by Yang and colleagues presents interesting data identifying mature regulatory dendritic cells (mregDCs) as a prognostic signature for predicting responses to immune checkpoint inhibitors (ICIs). The approach of using single-cell RNA sequencing for biomarker discovery provides a robust platform to assess the cellular composition of patient tumors and the potential role each cell type plays in influencing response to ICI. This dataset is substantial, comprising a large number of cells. The inclusion of single-cell ATAC sequencing adds valuable depth to the analyses. Additionally, the identification of key gene signatures distinguishing dendritic cell subtypes and the origin of mregDCs is a notable contribution. Overall the potential role of mregDCs in predicting response to ICI through orchestrating the tumor microenvironment is an interesting and original contribution. Furthermore the scRNAseq dataset will be a very valuable resource for future studies from other investigators.

Major comments

- The authors have presented data supporting mregDCs as a predictor of response to ICI. However, the manuscript could be strengthened by addressing how mregDCs may predict response to anti-PD1 monotherapy versus the combination of anti-CTLA4 + anti-PD1. While the sample size is limited, with only a total of 5 responders to ICI, for combination therapy the analyses are limited to only a single responding case to anti-CTLA4 + anti-PD1. Samples from patients treated with other combination therapies that include non-ICI therapies are potentially confounded with respect to understanding response to ICIs.

- Following the discovery analyses with scRNAseq the authors utilize bulk RNA-seq data to provides an additional layer and validation to the analyses. However I found the small biological differences in the median values raise some concerns about the robustness of the findings. While statistical significance is observed, the lack of notable biological differences limits the full impact of this result. Alternative single cell approaches to analyse mregDCs such as flow cytometry or CyTOF might more robustly address the interesting hypothesis.

- The classification of "high" and "low" signatures for the subtypes presents an intriguing method for stratifying patient responses. However, the thresholds used to define these categories are not clearly defined in the manuscript. Providing a more detailed rationale for the cutoff points and the biological significance behind them would clarify this methodology and make the findings more transparent. This clarification would also enable readers to better understand the criteria for response prediction.

Other comments

- The integration of previous single-cell RNA-seq data is one strength of this study in the supplemental data. The inclusion of cDC populations in those analyses would offers a valuable opportunity to expand the study's findings. Investigating whether similar predictive signatures are present in cDC populations, and assessing their potential in predicting ICI responses, could significantly enrich the manuscript. This would also provide a more comprehensive view of the immune landscape and its role in therapy response.

Minor comments

- The authors should provide citations for the gene signatures used to annotate immune cell types. This would strengthen their argument and increase the impact of the resulting atlas.

Version 2:

Reviewer comments:

Reviewer #1

(Remarks to the Author)

The authors have nicely addressed the previous comments and concerns. The manuscript is recommended for publication.

Reviewer #2

(Remarks to the Author)

The Authors have addressed the Reviewers' comments and we do not have any additional concerns.

Reviewer #3

(Remarks to the Author)

Reviewer #4

(Remarks to the Author)

While the authors addressed most of concerns raised, the way that authors handled the ECM-mregDC still lacks the necessary rigor. There are many supervised cell type inference tools for spatial sequencing data out there, and the authors should do their due diligence to explicitly locate mregDC and different fibroblast subsets (iCAF/mCAF), followed by formal neighborhood analysis to confirm statistically significant co-localization.

Reviewer #5

(Remarks to the Author)

The authors have addressed all important issues adequately with new analyses and revisions of the manuscript. The addition of spatial transcriptomic data, multivariate analyses, and additional analyses of treatment subsets in the validation cohort that in particular found association of the mregDC signatures with anti-CTLA4 therapy combined or in sequence with anti-PD-1, significantly strengthens the manuscript. The association with prior or combined anti-CTLA-4 therapy is notable and should be specifically referred to in the abstract.

Version 3:

Reviewer comments:

Reviewer #4

(Remarks to the Author)

The authors have adequately addressed the reviewer's comments and I have no further concerns.

REVIEWER COMMENTS

Reviewer #1, expertise in dendritic cells and anti-tumour immunity (Remarks to the Author):

Using scRNA-seq and snATAC-seq data, the authors have characterized the immune, non-immune, and tumor cell diversity of metastatic melanoma from patients, including 22 patients treated with immune checkpoint inhibitors (ICIs) and 3 patients not treated with ICIs. The main finding of the study is the identification that mature dendritic cells enriched in immunoregulatory molecules (mregDCs) are enriched among conventional dendritic cells (cDCs) upon anti-PD-1 treatment response. A high proportion of mregDCs also correlates with better progression-free survival (PFS) in patients receiving combination therapy plus anti-PD-1, and the mregDC signature in ICI-treated bulk samples correlates with better PFS. The epigenomic landscape of the cDC to mregDC transition is also nicely described using snATAC-seq. Overall, the study is well done and helps to advance our understanding of specific components (cell types, cell states) that correlate with ICI response. The data from this ICI cohort will be of particular interest to the community. The demonstration that mregDC proportion or signature serves as a good prognostic factor for progression-free survival in ICI-treated melanoma could advance tailored immunotherapy interventions. However, there are still some issues that need to be addressed to better substantiate the claims. The authors should carefully address all comments.

Major Comments

1. Analysis of Pre-, On-, and Post-Treatment Biopsies. Although the authors have used pre-, on-, and post-treatment biopsy samples, they have not conducted comparative analyses between these time points. The authors should emphasize the immune population shifts resulting from the treatment by comparing these samples.

Response to Reviewer: Thank you very much for taking observation to this. We agree that conducting comparative analyses between samples of different time points would be beneficial to our study. Although our primary analysis did not examine longitudinal effects, we do recognize the effect of treatment stage in cell-type composition which might influence clinical outcome. To address this, we have performed analysis highlighting immune composition by time point as well as by sample categorized by time point, and observed a trend of a shift from predominantly innate immune cell types to adaptive immune cells, such as CD8+ T cells, in both absolute numbers and proportions within the immune compartment (**Extended Data Fig. 3A**). We added the corresponding text in **lines 229-233**, *“To account for the limited sample size, we combined tumors collected at different treatment timepoints to identify cell types and transcriptomic features consistently enriched in responders and non-responders, irrespective of treatment stage, while acknowledging that treatment itself may influence tumor transcriptome profiles and cell composition (**Extended Data Figure 3A**)”*.

Pre-Treatment Biopsies

On-Treatment Biopsies

Post-Treatment Biopsies

2. Definition of Immune Cell Diversity (Figure 2).

- Myeloid and T Cells (Figures 2C, D, E): The clusters should be annotated with marker genes, including their average expression levels and possibly the number of unique molecular identifiers (UMIs) per cell. This information would be particularly useful for distinguishing macrophage clusters (see comment below) and differentiating between natural killer T (NKT) cells and certain T cell subtypes that are challenging to separate.

Response to Reviewer: Markers for the different immune cell subtypes are shown in **Extended Data Fig. 2C-F**, the number of UMIs per cell for these subtypes as shown below has been added to **Supplementary Fig. 2D**. As suggested by the reviewer, we observed higher UMI counts for NKT cells, particularly CD8 NKT cells, compared to other T cells, consistent with previous reports^{1,2}. These findings further validate our annotations. We thank the reviewer for their insightful comment. Relevant text has been added to **lines 152-154**, *“All CD8+ T cell subtypes showed a similar number of UMIs per cell, except for the NKT subtype, which had a higher UMI count as previously reported (Supplementary Fig. 2D)”*.

- Macrophage Compartment: The macrophage compartment cannot be transcriptionally separated into M1/M2 subtypes, as this is an *in vitro* dichotomy. The authors should refine their clustering using up-to-date definitions, such as resident-like macrophages, TREM2+ macrophages, and different subtypes of tumor-associated macrophages (TAMs) as recently described in pancreatic cancer (Dunsmore et al.). The lack of proper definitions could influence the study's findings. Similarly, the diversity of B cell populations should be increased if possible.

Response to Reviewer: We appreciate the reviewer's critique of the M1/M2 paradigm and agree that macrophage states *in vivo* are more nuanced. While retaining M1/M2 terminology for consistency with inflammatory (CXCL10^{high}/CD47^{high}) vs. anti-inflammatory (CD163⁺/TREM2^{high}) functional phenotypes³, we re-evaluated our clusters using recent TAM definitions⁴. Our 'M1-like' macrophages align transcriptionally with inflammatory/reactive macrophages (e.g., CXCL10^{high}), while 'M2-like' cells mirror TREM2+ immunosuppressive macrophages described in melanoma and other cancers (Figure a)⁵⁻⁷. Notably, our TAM clusters, not M1 and M2 clusters, share features with Dunsmore *et al.*'s HuTAMs B/C (Figure b). Importantly, regardless of nomenclature, anti-inflammatory macrophages (TREM2^{high}/CD163^{high}) remained enriched in non-responders, underscoring the robustness of our findings (Fig. 3A). We have added text clarifying these parallels (lines 143-148) and caveats about evolving macrophage taxonomy, "While we employ the M1/M2 nomenclature to denote inflammatory and anti-inflammatory macrophage states, these clusters align with *in vivo* macrophage subtypes described in melanoma and other cancers²³. For example, 'M2-like' macrophages resemble TREM2+ immunosuppressive macrophages and TAMs share transcriptional overlap with HuTAMs B/C identified in pancreatic cancer²⁴. These parallels highlight conserved functional states across malignancies, though evolving macrophage taxonomy underscores the need for standardized, context-specific definitions".

Figure a

Figure b

We sincerely appreciate the reviewer’s suggestion to deepen our characterization of B cell diversity. In response, we showed marker gene expression patterns for the B cell subtypes (new dot plot, **Extended Data Fig. 2F**) and integrated insights from two pan-cancer B cell atlases (Ma *et al.*, 2024⁸; Yu *et al.*, 2024⁹). Our analysis confirmed: 1) Plasma B cells (high CD27, PRDM1) align with Ma *et al.*’s Plasmocytes and MZB1+ Antigen-Secreting Cells (**Figure c**). 2) Memory B cells are CD27+IGHG+, consistent with Yu *et al.* (**Figure d**). While our dataset lacks the resolution to define all atlas-reported subtypes, these refinements confirm robust alignment with conserved B cell states across cancers. We have updated the text (**lines 185-189**) to clarify these parallels and limitations in subtype granularity due to sample size.

Figure c

Figure d

- Function Inference from Gene Expression: Anti-tumor or pro-tumor functions cannot be inferred solely from gene expression within a cluster or from the enrichment of a cluster itself. Such assertions should be removed throughout the paper. Only correlations with response, overall survival (OS), or progression-free survival (PFS) should be presented.

Response to Reviewer: We agree with the reviewer that cell functions are more complex than currently understood and have removed functional inferences based solely on gene expression profiles from **Fig. 2, Extended Data Fig. 2, Supplementary Fig. 3**, and the Abstract section (**lines 28-31**), the Results section “Cell subtype correlations reveal six modules” (**lines 200-216**) and “Mature and migratory cDC abundance associated with ICI response and PFS” (**line 235**) and the Discussion section (**lines 464-466,477**).

- Correlation of Cell Subtype Proportions: Correlations of cell subtype proportions should be performed on immune cells only. Introducing tumor cell subtypes can drive the correlation and potentially mask the immune archetypes that could be observed. Figure 2F shows that the correlations of cell subtypes are completely different in another cohort; performing immune-only correlations of cell subtype proportions could help clarify this. If the correlations remain different, the authors should discuss these discrepancies.

Response to Reviewer: We thank the reviewer for this thoughtful suggestion. To address whether tumor/stromal cells mask immune-specific correlations, we performed immune-only analyses and identified two conserved modules (**Figure a**):

1. Module 1: mregDCs correlate with CD8+/CD4+ naïve T cells, activated T cells, and naïve B cells.
2. Module 2: Tregs correlate with plasma B cells and exhausted CD8+ T cells.

While other modules differed between immune-only and full cohort analyses, we retained the original framework because tumor/stromal cells (e.g., via ECM pathways) directly influence immune cell abundance, as shown by the correlation between mregDCs and tumor translation initiation programs (**Extended Data Fig. 6E**). Critically, Module 1 remained conserved across cohorts, even when excluding tumor/stromal cells (**Figure b**), underscoring its biological robustness.

We acknowledge dataset-specific variability in other modules and have expanded the Results section (**lines 225-226**) to caution against overinterpretation while emphasizing the value of holistic tumor microenvironment profiling. These revisions strengthen our conclusions by balancing immune-centric insights with the reality of cross-compartment interactions.

Figure a.

Figure b.

3. Association of mregDC with Response (Figure 3):

- For all the clusters identified in Figures 1 and 2, the authors should present in a single figure which clusters are associated with response, which are associated with non-response, or which are not significant.

Response to Reviewer: Thank you for this comment. In our original submission, we chose only to show proportions within a select treatment group and associated with either response or non-response at a p-value ≤ 0.1 . We have expanded this analysis to include all subtype proportions detailed in Figures 1 and 2, grouping them by association with response or non-response at a p-value ≤ 0.1 , or not significant (p-value > 0.1), which is added as a new **Supplementary Fig. 5** and cited in the corresponding text (**lines 235,240**).

- Regarding the observed enrichment of plasma cells in non-responders instead of responders, the authors should discuss the accumulating evidence linking plasma cells and response to checkpoint blockade, such as in the study by Helmink et al. (Nature, 2019; <https://www.nature.com/articles/s41586-019-1922-8>).

Response to Reviewer: This is an interesting point. The plasma B cells identified in this study expressed high levels of IGHA1 and IGHA2 (**Extended Data Fig. 2F**), indicating they are likely IgA+ plasma cells. These cells have been reported to express IL-10 and PD-L1, with their presence linked to TGF β receptor signaling, suggesting an immunosuppressive role in the tumor microenvironment of prostate cancer³. Conversely, tumor-antigen-specific and antigen-independent IgA responses have been shown to inhibit ovarian cancer growth by coordinating tumor, T cell, and B cell interactions⁴. In the study by Helmink et al., an increase in plasma cells was observed in responders compared to non-responders in a neoadjuvant ICI setting, though this was not statistically significant and was largely influenced by data from a single patient in a CyTOF dataset of melanoma patients⁵. Notably, IgA was not explicitly reported in their analysis. Given these mixed findings, we believe the role of plasma cells in immunotherapy is complex and context-dependent, warranting further investigation. We have added this discussion in **lines 236-239**, "*The plasma B cells identified in this study expressed high levels of IGHA1 and IGHA2 (Extended Data Fig. 2F), suggesting they are likely IgA+ plasma cells. While IgA+ plasma cells have been reported as immunosuppressive in various cancer types, their role in the context of ICI therapy remains unclear*".

- Kaplan-Meier Analysis (Figures 3B and 3C). Please ensure that the Kaplan-Meier graphs are computed per patient rather than per sample.

Response to Reviewer: Thank you for raising this important detail. One unique characteristic of our data is that overall survival and progression-free survival are sample-level variables; that is, samples from the same patient have different OS and PFS because they were collected at different time points. Therefore, our initial Kaplan-Meier analysis included all samples. However; that does not mitigate the fact that samples are not independent because they may be from the same patient, and so we chose to revise our Kaplan-Meier to include one sample per patient of the latest time point (see figure below). We apply the same selection when running Cox regression (see response to **Detailed Points 8 for Reviewer #2 on pages 18-19**). Although our sample size decreases, we still observe similar trends in survival across various treatment group subsets. We also included more survival analysis for anti-PD1 only treated-samples to address **Major Comment 1 from Reviewer #5 on pages**

26-27. We have added the figure to **Supplementary Fig. 6D** and this discussion to **lines 261-267**, “*We conducted multivariate Cox regression analysis to address dataset heterogeneity and confirmed that both continuous and binarized mregDC proportions are significant predictors of PFS in ICI-treated samples. To account for non-independence of longitudinal samples from the same patient, survival analyses (Kaplan-Meier and Cox regression) were revised to include only the latest timepoint per patient, reducing the cohort to n = 11 samples. Despite this adjustment, trends in survival benefit for mregDC-high tumors persisted (Supplementary Fig. 6D), underscoring the robustness of our findings.*”

- mregDC Gene Signature (Figures 3D and 3E): It is unclear what mregDC gene signature was used, even after examining Supplementary Table 3, "Non-tumor cell subtype markers in reference to all the other cells." We do not know the cutoff in terms of the number of marker genes used to define the mregDC signature. This information is crucial if this signature is to be used in future predictions of response to immune checkpoint blockade. The signature and the number of genes used should be clearly presented in the text, figures, or tables.

Response to Reviewer: We agree with the reviewer on the importance of clarifying the gene signature. The signature includes 886 genes, each with an adjusted p-value <0.05 and an average log2 fold change >0 when comparing mregDCs to all other cells in the tumor microenvironment. To enhance clarity, we have added a column to **Supplementary Table 3** indicating whether each gene is part of the signature and clarified this threshold in the corresponding text (**lines 280-281**).

4. Transcriptional Landscape of cDCs

- The authors assert that mregDCs express higher levels of IRF4 compared to cDC2. This should be validated at the protein level. This can be achieved by using cryopreserved cells from the cohort and performing intracellular staining with an anti-IRF4 antibody, comparing DC1 CD141+ CCR7+ (mregDC) and – (non mregDC) and DC2 CD1c+ CCR7+ (mregDC) and – (non mregDC).

Response to Reviewer: We sincerely thank the reviewer for highlighting the importance of protein-level validation. In response, we performed multiplex immunofluorescence staining on FFPE sentinel lymph node tissues from metastatic melanoma patients, which revealed that CD11C+CCR7+ cells (enriched in mregDCs) either co-express IRF4 and IRF8 or predominantly express IRF8 alone (**Fig. 4C**). While we were unable to stratify by DC1 (CD141+) and DC2 (CD1c+) subsets due to technical constraints in FFPE-based multiplexing, our data robustly demonstrate that CCR7+ mregDCs—regardless of subset—express IRF8 and/or IRF4 at the protein level. Future studies using flow cytometry on cryopreserved samples will further resolve subset-specific IRF4 dynamics. We have

added these results to **Fig. 4C** and expanded the Results section (**lines 300-301**), "*Multiplex immunofluorescence staining of sentinel lymph node tissues from metastatic melanoma patients confirmed protein-level co-expression of IRF4 with IRF8 in CD11C+CCR7+ mregDCs (Fig. 4C)*", and Methods section (**lines 968-983**) to detail the staining protocol.

- It cannot be concluded from the current data that most melanoma mregDCs are derived from DC1. Demonstrating this would require fate mapping experiments in mice or lineage tracing in patient-derived organoids (which we are not requested for this revision). The authors should modify this claim or discuss. Previous CITE-seq experiments in non-small cell lung cancer and hepatocellular carcinoma (Maier et al. Magen et al.) concluded mregDCs originate from both DC1 and DC2, but that likely more DC2 based on CD1c versus CD141.

Response to Reviewer: We acknowledge the controversy raised by the reviewer and have revised the relevant statements (**line 326**), added an additional reference (Magen et al) in **line 516**, and included further discussion in **lines 511-515**, as follows: "*Alternatively, cDC2s may undergo more extensive epigenomic reprogramming during maturation, contributing to their plasticity in the TME. Recent studies have also indicated that a larger proportion of mregDCs may originate from cDC2, based on staining for CD141 (a cDC1 marker) and CD1c (a cDC2 marker). Determining the precise lineage of mregDCs in vivo will require lineage-tracing experiments.*"

5. Epigenomic Landscape (Figure 5)

- Given the 43,499 single nuclei obtained, they should provide an epigenomic atlas of the other populations, not just the cDCs. They can show epigenetic changes within T cells (e.g., during exhaustion) and other myeloid cells, such as monocytes and macrophages. It would also help to define the changes in accessible chromatin regions between responders and non-responders. Furthermore, presenting the epigenomic landscape across the populations co-enriched with mregDCs in responders versus non-responders would make for a better Figure 6 than the current one, which presents findings that are difficult to leverage in terms of basic or clinical significance.

Response to Reviewer: We appreciate the reviewer's suggestion to extend our snATAC-seq analysis to additional populations and agree that such an epigenomic atlas would offer valuable

insights. However, our study focuses on cDCs, whose scRNA-seq and snATAC-seq data have been rigorously validated to ensure robust findings. Expanding the analysis to all populations in the ATAC space, while worthwhile, would require substantial time for proper validation and exceed the scope of this manuscript. Including incomplete analyses risks diluting the clarity and impact of our cDC-focused narrative. Additionally, the limited number of cDCs in our ATAC dataset, combined with the need to account for sample-level covariates, poses challenges for defining chromatin accessibility changes between responders and non-responders. Our current analyses provide actionable insights without overburdening readers. Future efforts will focus on generating a broader epigenomic atlas of other populations and enriching for cDCs from clinical samples to examine their epigenomic differences in responders versus non-responders.

Minor Comments

- The introduction includes a list of previously published scRNA-seq studies, which is unnecessary given that Supplementary Figure 1D already provides a comparison. Instead, the authors should focus on providing conceptual points that introduce the major findings of the study, such as details on the dendritic cell epigenome or background information on TCF1⁺ CD8⁺ T cells.

Response to Reviewer: We have updated the introduction to highlight the conceptual gaps addressed by this manuscript, as follows in **lines 69-76**: *“Due to limitations in cell collection and technology costs, these studies have primarily focused on tumor and immune cells, particularly T cells. In addition to the number and spatial distribution of T cells, which predict clinical response to ICIs, the presence of TCF7 protein in CD8+ T cells has been shown to be critical¹⁰. These stem-like TCF7+ CD8+ T cells maintain their stemness by expressing adult stem cell genes¹¹. T cell factor 1 (TCF1, encoded by TCF7) has been reported to optimize tumor-specific CD8+ T cell priming in low-antigenic environments¹². However, the role of myeloid cells, particularly dendritic cells, in influencing T cell states such as TCF7+ CD8+ T cells in human tumors in vivo remains unclear.”* and in **lines 85-88**: *“Although prior studies associated a higher density of mregDC-like dendritic cells with improved survival in cutaneous melanoma¹³, this state remains poorly characterized in terms of its epigenomic profile and interactions with T cells compared to other DC populations, especially in a clinical treatment setting.”*

- Clarification on cDCs in Immune Surveillance: The statement, "While immature cDCs are essential for immune surveillance, their maturation is vital for immunogenicity involving antigen processing, presentation, and T-cell co-stimulation," requires correction. The role of cDCs in immunosurveillance is predominantly attributable to mature cDCs.

Response to Reviewer: Thank you for highlighting this oversight. We have revised the sentence to read: *“While immature cDCs are essential for antigen capture and processing, their maturation is critical for immune surveillance through antigen presentation and T-cell co-stimulation”* (**lines 80-82**).

- Relevant Citation: When mentioning that “Although prior studies associated a higher density of mregDC-like dendritic cells with improved survival in cutaneous melanoma, this state remains poorly characterized in melanoma across different treatments.” the authors should cite: Barry KC, et al (PMID:

29942093), which demonstrates the positive prognosis of CCR7⁺ DC1 (mregDCs originating from DC1) in melanoma during PD-1 immunotherapy.

Response to Reviewer: We have added the citation (line 86).

- The table in Figure 1A should be simplified to present data per patient rather than per sample to enhance clarity.

Response to Reviewer: Thank you for your suggestion. The table in Fig. 1A is consolidated per sample as we considered this to be a good way of displaying all pertinent data in one figure, as only a few metrics such as age and sex are applicable on the wider per patient axis. Metrics like response, progression-free survival and overall survival, are all lesion specific. However, we have separated this figure into two tables to distinguish per patient metrics from those that are per sample here.

- In figure 1B The treatment group, treatment state, and lesional response information should be moved to the supplementary material.

Response to Reviewer: Thank you for the suggestion. We have incorporated the adjustments in Figure 1B and Extended Figure 1 accordingly.

- Given that the study primarily focuses on immune clusters, Figures 2A and 2B should be placed in the supplementary section.

Response to Reviewer: We agree with the reviewer and have moved the two figure panels into Extended Figure 2.

- In Figure 3 and Associated Extended Figures the proportions of mregDCs and cDCs should also be depicted among all MNPs to provide a comprehensive view of their distribution.

Response to Reviewer: We agree with the reviewer that these proportions offer a more comprehensive view of mregDC and cDC distributions. Our analysis showed no significant differences in these proportions between responding and non-responding tumors, emphasizing the relevance of the

relative proportion of mregDC within cDC. The following figures have been added as new panels in **Supplementary Fig. 6B,C**, along with corresponding text (lines 251-253).

- The statement, "mregDCs [...] play an important role in mediating ICI effects in mouse models²⁷," should be rephrased. The referenced study primarily demonstrates the modulation of PD-L1-expressing CD11c⁺ cells. While it is plausible that mregDCs play a role in this context, this has not been formally proven in that particular paper.

Response to Reviewer: We have revised the statement to: "*PD-L1-expressing DCs (mregDCs, highly expressing PD-L1) were shown as key mediators of ICI effects in mouse models*" (lines 246-247).

- It should also be noted that both human and mouse mregDCs express CCL22 and CCL17, as reported in the literature—not just mouse mregDCs.

Response to Reviewer: We sincerely appreciate the reviewer's attention to detail. To clarify, our original statement aimed to highlight two key points: 1) Both human and mouse mregDCs express higher levels of CCL17 and CCL22 compared to cDC1 and cDC2. 2) In humans specifically, cDC2s exhibit uniquely elevated expression of STAT6 compared to both cDC1s and mregDCs—a distinction not observed in mice¹⁴. We recognize that our original wording ("in contrast to mouse mregDCs") could imply a cross-species comparison of mregDC-specific markers, which was not our intent. To resolve ambiguity, we have revised the text (304-308) to emphasize intra-species differences: "*Similar to mouse mregDCs, human mregDCs express elevated levels of IL4I1, CCL17, CCL22, TNFRSF4, and BCL2L1 compared to human cDC1 and cDC2 subsets. However, human cDC2s uniquely exhibit higher expression of STAT6 compared to both human cDC1s and mregDCs (Supplementary Fig. 7B).*" This revision underscores that the distinction lies in human cDC2-specific markers rather than a direct species-to-species contrast.

Reviewer #2, expertise in single cell multi-omics and melanoma immunotherapy (Remarks to the Author):

In this manuscript, the authors aim to delineate the complex landscape of the tumor microenvironment (TME) in metastatic melanomas. They performed single-cell RNA sequencing on approximately 189,000 cells across 36 samples and single-nucleus transposase-accessible chromatin sequencing on around 43,000 cells from 14 samples in a metastatic melanoma cohort. They identified 55 immune, tumor, and stromal subtypes, including a co-enriched population module consisting of mregDC and naive T and B cells. They found that mregDC abundance was associated with better progression-free survival (PFS) in ICI-treated patients and a potential synergistic effect with the TCF7+/- CD8 T cell ratio in predicting patient survival. Subsequently, they performed cell type/subtype-specific gene, pathway, immune response enrichment, and ligand-receptor interaction analyses for mregDCs, suggesting their roles in enhancing the immune response against cancer.

Major Concerns:

While this dataset is valuable and the authors tried to address an important knowledge gap in the cancer immunity related to mregDCs, the analysis lacks depth. Major revisions are needed to integrate or connect different parts of their analysis, such as tumor co-expression module analysis, cell subtype co-enriched module analysis, DEG, interactome and pathway analysis, and snATAC analysis of mregDC vs cDC1/cDC2. Additionally, the survival analysis lacks necessary controls for many confounding factors, casting doubt on the robustness of their conclusions regarding mregDC as a prognostic predictor of patient survival.

Detailed Points:

1. In Figure 2, the authors assigned tumor cells to 15 pan-cancer meta-programs previously compiled. However, since this is a melanoma dataset, they should also consider conventional melanocytic vs. dedifferentiated subtypes, the latter being resistant to targeted therapy and immunotherapy (<https://doi.org/10.1016/j.ccell.2018.03.017>).

Response to Reviewer: We thank the reviewer for this important suggestion. Our analysis indeed aligns with melanoma subtype classifications: while most tumor meta-programs expressed high MITF (indicative of melanocytic differentiation, **Figure a**), two programs—Respiration and Stress—showed elevated dedifferentiation markers. The Respiration program (AXL^{high}/SMAD3^{low}/SOX9^{low}) resembles an undifferentiated subtype, while the Stress program (AXL^{high}/SMAD3^{high}/SOX9^{high}/EGFR^{high}/NGFR^{high}) reflects a neural crest-like state (**Figure b**), both of which were enriched in non-responders (**Fig. 3A, Extended Fig. 3B**). This mirrors prior work (Tirosh et al., 2016¹⁵) linking dedifferentiation to therapy resistance, underscoring the clinical relevance of our meta-programs in stratifying melanoma biology and treatment outcomes. We added the following figures to **Supplementary Fig. 2B,C** and corresponding texts in **lines 128-133**, *“While most tumor meta-programs exhibited melanocytic differentiation (high MITF and CTNNB1), two meta-programs—Respiration and Stress—expressed dedifferentiation markers including AXL, SOX9, EGFR and NGFR (Supplementary Fig. 2B,C). The Respiration program resembles an undifferentiated subtype, while the Stress program reflects a neural crest-like state. Both were later enriched in non-responders, aligning with the known resistance of dedifferentiated melanomas to targeted therapy and immunotherapy.”*

Figure a

Figure b

2. Cell Subclustering Analysis: How is this TAM distinct from M1 and M2 (Figure 2C)? It expresses classical M2 markers such as MRC1 and CD163. Additionally, markers mentioned in the manuscript, like VEGFA, IL-10, TNF, and TGFB1, were not shown in Extended Figure 2B. For B cells, why is the plasma cell called regulatory?

Response to Reviewer: We thank the reviewer for these critical points. Below, we clarify each concern: 1) TAM vs. M1/M2 Distinction: While TAMs express canonical M2 markers (MRC1, CD163), their transcriptional profile includes tumor-specific features (e.g., TREM2) and aligns with TAM subtype signatures from other cancers (Dunsmore *et al.*⁴), distinguishing them from classically defined M2 macrophages. These nuances are now detailed in the revised text (lines 143-148) and visualized in Extended Data Fig. 2C. 2) We have revised the text (lines 142-143) to align with the markers shown in the figure, ensuring clarity and accuracy. All transcriptional markers used to define TAMs—including those validated against recent TAM taxonomies (e.g., TREM2, SPP1, CD163)—are listed in Supplementary Table 3, which now serves as a centralized reference for reproducibility and cross-study comparison. 3) Plasma Cell Annotation: Our plasma cell cluster is defined by canonical markers (CD27, PRDM1) and lacks regulatory B cell markers (CD1D). The term 'regulatory' was inadvertently

misapplied; this has been corrected in the text (**line 188**) to avoid confusion. We appreciate the opportunity to improve clarity and rigor in our annotations.

3. MALAT1 can indicate low-quality cells, so the authors should be cautious about their definition of HNRNPH1+ Teff cells in Fig 2C (<https://kb.10xgenomics.com/hc/en-us/articles/360004729092-Why-do-I-see-high-levels-of-Malat1-in-my-gene-expression-data>).

The same check should be done on the high level of heat shock protein - is this real subset or artifact of single cell RNAseq?

Response to Reviewer: The issue raised regarding MALAT1 and heat shock protein as a potential indicator for low quality cells is an important one. Given that these markers are of importance within the context of the findings discussed in this paper, a set of markers for dying and dead cells from Rich et al., 2024¹⁶ were utilized as a way of discerning whether expression of the aforementioned genes relates to issues of cell quality. Results returned no discernable evidence of low quality cells within the cluster the reviewer highlighted as a concern, namely the HNRNPH1+ and Tex/HS clusters. Given this, it is a reasonable conclusion that the expression of these genes are of relevance to the analysis discussed in the paper, and not an artifact. The following figure has been added as **Supplementary Figure 2E** and the relevant text has been inserted in **lines 158-160**, "*To confirm that this subtype was not the result of low-quality cells, we scored all CD8+ T cells using a low-quality cell signature28 and found comparable quality levels across all cells (Supplementary Fig. 2E).*"

4. Correlations of cell subtype proportions (Figure 2F): The annotation of the modules is dubious. For example, why is module 1 called "Anti-tumor immune infiltration"? This is misleading because the module represents groups of subtypes correlated in their relative proportion across samples; there is no immunofluorescence data showing that the immune subtypes in module 1 specifically infiltrate the tumor parenchyma. Module 6 seems to have more tumor-infiltrating features with multiple effector T cell subtypes and interferon response in the tumor population. Also, how is module 4 an anti-tumor module with multiple proliferating tumor cell populations?

Response to Reviewer: We agree with the reviewer that the module annotations were overstated. We have removed the questionable inferences from **Figure 2, Extended Data Figure 2,**

Supplementary Figure 3, and the corresponding text (lines 203-216). Thank you for bringing this to our attention.

5. The statement of “conserved cell subtype modules regardless of treatment” (page 4, Figure 2F, Supplementary Figure 2A) is questionable. Among the 36 samples, 30 were treated with ICI, so it is not surprising that the overall correlation modules pattern is dominated by ICI-treated samples. Indeed, Supp Fig 2C looks quite different (is this atlas analysis include Sade Feldman's?) The authors should perform separate module analyses for ICI-untreated and -treated samples to see if the correlation module pattern changes.

Response to Reviewer: We sincerely thank the reviewer for this critical insight. In response, we performed separate module analyses for pre-treatment vs. on/post-treatment samples (both below and now **Supplementary Fig. 3B,C**), which revealed distinct correlation patterns. Pre-treatment samples exhibited unique modules (e.g., mregDC correlated with TAM, stress and hypoxia tumor programs and inflammatory fibroblast), consistent with its immunoregulatory phenotype, reflecting baseline immune states. On/post-treatment samples mirrored the combined analysis pattern (e.g., mregDC correlated with CD4 and CD8 naive cells, CD8 effector T cells and the translation initiation tumor program), suggesting therapy-driven convergence. This confirms that treatment status significantly shapes module interactions, and we have revised the original statement (lines 219-222) to clarify that conservation is context-dependent, not absolute, “To assess treatment-dependent module dynamics, we performed separate analyses for pre-treatment and on/post-treatment samples. While on/post-treatment samples recapitulated the combined cohort’s module patterns, pre-treatment samples exhibited distinct correlations (Supplementary Fig. 3B), revealing that module conservation is context-dependent and shaped by therapy.”

Regarding Sade Feldman’s dataset, it was excluded because it profiles only immune cells, whereas our module analysis requires all tumor microenvironment cell types for holistic interpretation. We agree that expanding this framework to immune-focused datasets is a valuable future direction. Please see **page 5** for related analyses.

Pre-treatment

On/Post-treatment

6. Association between mRegDC and ICI response. Did the authors compare the differences in monocyte, macrophage and DCs among R and NR to ICI against Sade Feldman Cell 2018 dataset?

Response to Reviewer: We appreciate the suggestion to take advantage of external datasets, such as the Sade Feldman Cell 2018 dataset, to validate our findings of the association between myeloid subtypes and ICI response. All patients in the Sade-Feldman dataset were treated with ICI. Using the cell type annotation through label transfer, we further looked into the distribution of macrophages, monocytes, and conventional dendritic cells (cDCs). Only 44 cDCs were identified in the entire dataset, with most samples having only 1-2 cDCs, and it would be extremely difficult to find significant associations between cDC subtype proportions and response. However, there was a greater population of monocytes and macrophages which an association analysis could be conducted on.

Among monocyte and macrophage subtypes, M1 macrophages, tumor-associated macrophages, and nonclassical monocytes showed similar proportional trends across both datasets but without statistical significance. Classical monocytes were more abundant in responding samples in both datasets but reached statistical significance only in ours. M2 macrophages, associated with anti-inflammatory activity, were significantly higher ($p < 0.1$) in both datasets, though more so in responding samples from Sade Feldman, which contrasts with their expected enrichment in non-responding samples as shown in our dataset. However, both datasets have relatively small sample sizes (ours: 5 responders, 13 non-responders; Sade Feldman: 4 responders, 16 non-responders treated with ICI), limiting definitive conclusions about M2 macrophages. Despite this, our findings, while preliminary, provide a foundation for further investigation, particularly focusing on the mregDC subtype and its clinical relevance.

7. Survival analysis in Figures 3, Extended Figure 3 and Supplementary Figure 3: This section raises several concerns. The authors compared the relative fraction of different cell subtypes between ICI responders (R) and non-responders (NR). However, among the 19 ICI-treated samples, 7 are pre-treatment, 4 are on-treatment, and 5 are post-treatment (are these ICI resistant?). Combining pre- and post-treatment samples for comparison is questionable because the treatment itself could have altered the tumor transcriptome profile and cell type composition.

Response to reviewer: We sincerely thank the reviewer for raising this critical point. While we recognize that treatment may dynamically alter tumor transcriptomes and immune composition, our analysis intentionally combined samples across timepoints to identify biomarkers persistently associated with response or resistance, irrespective of treatment stage. Though limited sample size precluded robust timepoint-specific stratification, this approach increases statistical power to detect stable biological signals that transcend transient therapy-induced fluctuations. Importantly, responding and non-responding tumors were distributed across all timepoints (**Fig. 1A**), and we validated the key finding, mregDC enrichment in responders, remained consistent when controlling for temporal variation (see response to your next comment). We have expanded our result discussion (**lines 230-233**) to explicitly acknowledge this limitation and underscore the need for larger longitudinal cohorts to disentangle therapy-driven vs. response-specific dynamics in future work.

8. For the survival analysis, beyond Kaplan-Meier plots, the authors should perform multivariate analysis to account for potential confounding clinical factors (e.g., treatment type, treatment state, days to last ICI treatment) to show whether mregDC is an independent prognostic predictor.

Response to reviewer: We agree with performing multivariate analysis to account for the heterogeneity within our dataset. We have previously applied multivariate analysis but did not show it in our figures. Using Cox-regression, we present two models by sample and by patient. Because samples were collected at different time points, overall survival and progression-free survival were recorded on a sample level. Hence, for our first model we decided to apply Cox regression on samples, similarly to how we applied Kaplan Meier analysis in our original submission. For the second model, we used the latest sample per patient, such that there was one sample per patient. The covariates chosen for each model are detailed below:

Model 1 (using samples) covariates	Model 2 (using one sample per patient) covariates
age	age
sex	sex
subtype, grouped into cutaneous or other	subtype, grouped into cutaneous or other
ICI status (none, with ICI, combo, PD1)	ICI status (none, with ICI, combo, PD1)
treatment state (PRE/ON/POST)	treatment state (PRE/ON/POST)

For both models, we tested the predictive power of mregDC in two forms: 1) mregDC/cDC proportion, and 2) mregDC/cDC status as defined throughout our manuscript, where a proportion less than 20% is low, any proportion greater than 20% is classified as high (see rationale under **Reviewer #5 Major Comment 3 on pages 28-29**).

For Model 1, we found mregDC proportion and status to be a significant predictor for progression-free survival (PFS) in many of the different treatment cohorts. The p-values for mregDC proportion and status can be found below for the sample PFS models.

Treatment group	mregDC/cDC p-value	mregDC/cDC coefficient	mregDC status p-value	mregDC status coefficient
All samples (n = 35)	0.0424	-4.053	0.0538	-1.116
Samples treated with some form of ICI (n = 29)	0.0332	-4.918	0.0183	-1.555
Samples treated with combo or anti-PD1 (n = 18)	0.0181	-9.313	0.0539	-1.920
Samples treated with anti-PD1 (n = 14)	0.0069	-13.19	0.0129	-3.205

For Model 2, one sample was chosen per patient. We chose the sample at the latest time point to represent a patient's status. Although mregDC proportion and status did not reach significance which is likely due to a decreased sample size, the coefficients were in the direction of better clinical outcome. As the sample size further decreased when focusing on treatment groups, the models started to fail to converge.

In summary, multivariate analysis confirmed that a high relative proportion of mregDC is significantly associated with improved PFS in ICI-treated samples. Corresponding text has been added in **lines 261-267**, "*We conducted multivariate Cox regression analysis to address dataset heterogeneity and confirmed that both continuous and binarized mregDC proportions are significant predictors of PFS in ICI-treated samples. To account for non-independence of longitudinal samples from the same patient, survival analyses (Kaplan-Meier and Cox regression) were revised to include only the latest timepoint per patient, reducing the cohort to n = 11 samples. Despite this adjustment, trends in survival benefit for mregDC-high tumors persisted (Supplementary Fig. 6D), underscoring the robustness of our findings.*"

9. Additionally, there are discrepancies in sample numbers between different figures. Figure 3A, B shows data from 18 combo or anti-PD1 samples, but Figure 1A shows there are 19 combo or anti-PD1 samples; Figure 3C shows a "total" of 33 samples, but the cohort is 36 samples.

Response to reviewer: We sincerely thank the reviewer for their meticulous attention to detail. The discrepancies in sample numbers across figures arise from context-specific exclusions required for rigorous analysis. **Fig. 3B** (18 vs. 19 samples): One anti-PD1/combo-treated sample lacked detectable mregDCs and was excluded from survival analysis to avoid bias from missing cell-type data. **Fig. 3C** (33 vs. 36 samples): Three samples lacked CD8+ T cells and were excluded from TCF7 analysis, as this metric requires CD8+ T cell presence for biologically meaningful interpretation. All sample inclusion/exclusion criteria are now explicitly detailed in figure legends (**Fig. 3B-C**) to enhance transparency.

10. Synergy Between mregDC and TCF7+/- CD8 T Ratio: One interesting finding is the potential

synergy between mregDC and TCF7+/- CD8 T ratio in predicting patient survival. I can only see "high" and "low" stratification. What is the definition of "high/low"? I would expect to see 4 strata here.

Response to reviewer: Thank you for observing this. We originally defined two strata, with "high" being mregDC high and TCF7+/- high, and "low" being all other samples. However, we agree that the "low" category should be further delineated by TCF7+/- ratio. There were three strata, as no samples had a high mregDC/cDC proportion and low TCF7+/- ratio. This is consistent with the correlation between the two signatures (**Extended Data Fig. 3E**).

We performed this analysis, using the sample with the latest timepoint if a patient had multiple samples. Although the stratification leads to fewer samples in each category, the patients whose sample had a high mregDC/cDC and TCF7+/- ratio generally had longer progression-free survival and overall survival, but did not reach significance (see below). We also explained the classification we used in the figure more clearly in the corresponding text (**lines 276-279**), "*samples with high mregDC proportion and high TCF7+ CD8 ratio had higher survival probability regardless of treatment type compared to samples with either low mregDC proportion or low TCF7+ CD8 ratio (Fig. 3C). Interestingly, no samples had a high mregDC/cDC proportion and low TCF7+/- ratio.*"

11. The authors should conduct more in-depth analysis of this angle. For example, do mregDC and TCF7+ CD8 T cells (which of the T cell population in Fig 4G is TCF7 high?) show heightened interactions of antigen presentation (cellchat), co-stimulatory pathway or other T cell/DC related pathways within the TME?

Response to reviewer: We appreciated the suggestion of looking into T cell/DC interactions specific to mregDC and TCF7+ CD8 T cells. We partitioned CD8 T cells to TCF7+ (>0 read count) and TCF7- T cells (0 read counts) and applied the same computational method, CellPhoneDB, to find potential cell-cell interactions that are differential to TCF7+/- CD8 T cells and mregDC/cDC1/cDC2. We again observed a potential CCL19-CCR7 and CD70-CD27 interactions between mregDCs and both TCF7+ and - CD8 T cells, as we have described in **Fig. 4H**. Additionally we observed a potential interaction in the co-stimulatory PVR-CD226 axis²⁰ specific to mregDC and TCF7+ CD8 T cells. However, we also observed a potential binding of HLA-E on mregDC to the co-inhibitory checkpoint receptor NKG2A (encoded by KLRC1 gene) on TCF7+ CD8 T cells. These results suggest the interaction between mregDC and TCF7+ CD8 T cells could have dynamic and complex functions on T cell activation. We also acknowledge that high sparsity of TCF7 expression in our scRNA-seq dataset could limit the power of this analysis in inferring TCF7+ CD8 T cell interactions.

12. The validation of mregDC signature's prognostic power in an independent bulk RNAseq dataset (Figure 3D) is not convincing because the mregDC signature score in bulk RNAseq could reflect overall immune infiltration, which is associated with better survival and ICI response. The authors should provide more detailed analysis to show whether other immune subtypes or overall immune infiltration levels (using tools like ESTIMATE) are also associated with longer PFS/OS.

Response to reviewer: We thank the reviewer for raising this critical point. While the mregDC signature correlates with overall immune infiltration (inferred using ESTIMATE immune score, **Extended Data Fig. 3I**), multivariate analysis revealed that the combination of mregDC and immune score significantly predicted progression-free survival ($p = 0.047$), though neither covariate reached individual significance. This suggests mregDC and immune infiltration capture complementary aspects of the tumor microenvironment, jointly enhancing prognostic power beyond either metric alone. Notably, prior clinical trials (e.g., NCT01976585) demonstrate that dendritic cell recruitment and activation drive intratumoral immune infiltration, positioning mregDCs as upstream regulators rather than passive bystanders. Our findings align with this model: mregDC-enriched tumors exhibit coordinated immune infiltration and activation (e.g., CD8+ T cell infiltration, IFN- γ signaling), which may amplify anti-tumor immunity. We have expanded the results (**lines 288-292**), *"While the mregDC signature correlated with overall immune infiltration (ESTIMATE immune score, $p < 2.2 \times 10^{-16}$; **Extended Data Fig. 3I**), multivariate analysis revealed that the combination of mregDC and immune score significantly predicted PFS ($p = 0.047$), though neither covariate reached individual significance. This suggests mregDC and immune infiltration capture complementary aspects of the TME, jointly enhancing prognostic power beyond either metric alone"*, and discussion (**lines 477-482**) to clarify this mechanistic interplay, *"Its significant correlation with overall immune infiltration as inferred by the ESTIMATE immune score in a meta-RNA-seq cohort aligns with clinical evidence that DC recruitment and activation drives immune infiltration (NCT01976585), suggesting mregDCs may orchestrate broader anti-tumor immunity rather than merely reflecting it. Prognostic power of the derived mregDC signature*

supports its role as a biomarker of functional DC engagement, which could guide strategies to amplify DC-mediated immune infiltration".

13. Comparison with Previous Findings of mregDC: As a data-analysis-based paper with no functional in vitro and in vivo validation, the authors should compare their findings with previous studies on cancer immunity and mregDC-T cell interactions (e.g., PMID: 38728412, 34343496, 37708889). For example, is there heightened CD40:CD40L stimulation of CD4 T cells by mregDC? How about antigen presentation or cross-presentation pathways? What about DC-derived chemokine signals (e.g., CXCL16, CXCL9, CXCL10, IL-12, IL-15)?

Response to reviewer: We sincerely thank the reviewer for prompting this critical comparison. Below, we contextualize our findings with the cited literature, integrating new data that refine the role of mregDCs in melanoma immunity: 1) CD40:CD40L and mregDC-T Cell Interactions. While CD40 is expressed on mregDCs (**Supplementary Fig. 7B**), its interaction with CD40L on T cells was not prominent in our dataset. This suggests that in our dataset mregDCs in responders may prioritize cytokine-driven (IL-12/IL-15) T cell activation and survival over canonical CD40:CD40L co-stimulation, as observed in Pilato *et al.* (2021). 2) Antigen Presentation and Cross-Presentation. We observed elevated MHC-I genes in mregDCs (**Supplementary Fig. 7D**), suggesting a potential role in antigen presentation to CD8+ T cells, akin to observations in Pilato *et al.* (2021). However, canonical cross-presentation genes (e.g., TAP1) remained highest in cDC1s (**Supplementary Fig. 7C**), aligning with their established role in MHC-I antigen processing (Pittet *et al.*, 2023). This implies that mregDCs may engage CD8+ T cells via MHC-I while relying on cDC1s for efficient cross-presentation. 3) DC-Derived Chemokine Signals. Our data reveal a human melanoma-specific chemokine division of labor: mregDCs produce CXCL9/10 and IL-12/IL-15 (**Fig. 4D,F; Supplementary Fig. 7C**), mirroring Pittet *et al.*'s (2023) model of chemokine-cytokine synergy in local T cell activation/survival. cDC2s uniquely express CXCL16 (**Supplementary Fig. 7B**), contrasting with Pilato *et al.*'s (2021) mouse model where CXCL16 marked mregDCs. This discrepancy may reflect species-specific DC programming or tumor microenvironment differences, highlighting the need for human-focused validations. We included these results in **lines 328-330**, "Furthermore, we compared MHC I and MHC II gene expression across the three cDC populations and found that MHC II genes were highest in cDC2, while MHC I genes were upregulated in mregDC, suggesting mregDCs primarily present antigens to CD8+ T cells via MHC I

(Supplementary Fig. 7D), discussion in **lines 515-519**, “A key finding of our study is the identification of an mregDC-specific enhancer regulating IL15 expression. This enhancer, absent in cDC1/cDC2 subsets, provides a transcriptional blueprint for the elevated IL-15 production observed in mregDCs. Prior work established IL-15 as a survival signal for tumor-infiltrating CD8+ T cells⁷⁴, but our data now link this cytokine to a stable epigenetic program unique to mregDCs”, and added a new figure panel **Supplementary Fig. 7D**.

14. snATAC-seq Analysis: The authors identified enriched TF motifs of five NF- κ B TF family members in mregDC. How do these results reconcile with the scRNAseq analysis? Among the DEGs identified in scRNAseq, do any show corresponding more accessible chromatin regions and correlation with TF motif enrichment in these regions?

Response to reviewer: We thank the reviewer for this insightful question. Our multi-omics approach demonstrates strong concordance between NF- κ B motif enrichment in accessible chromatin regions (snATAC-seq) and NF- κ B-driven transcriptional programs in mregDCs (scRNA-seq). SCENIC analysis of scRNA-seq data revealed that NF- κ B signaling regulons (e.g., NFKB1, REL and NFKB2) are significantly upregulated in mregDCs compared to cDC1/cDC2 (**Extended Fig. 4D**). These regulons encompass co-expressed NF- κ B target genes, confirming that NF- κ B TF activity is not only reflected in chromatin accessibility but also functionally drives transcriptional programs in mregDCs. Additionally, we identified six mregDC-specific enhancers, four out of the six harboring NF- κ B motifs in regions of open chromatin and one at the corresponding promoter (**Fig. 5E, Extended Fig. 5D**). These enhancers are linked to genes upregulated in mregDCs (IL15, FSCN1, DUSP22, KDM2B, GLS, CCR7), directly connecting NF- κ B motif accessibility to elevated expression. The synergy between NF- κ B motif accessibility (snATAC-seq), regulon activation (SCENIC/scRNA-seq), and enhancer-linked DEGs provides robust, multi-layered evidence that NF- κ B is a master regulator of mregDC identity. This consistency across datasets underscores the reliability of our findings. We have highlighted these results in **lines 403-404**, “This enrichment was validated by the previous SCENIC analysis, which identified upregulated NF- κ B regulons driving transcription (**Extended Fig. 4D**)”, and **412-414**, “These genes (IL15, FSCN1, DUSP22, KDM2B, GLS, and CCR7) are significantly upregulated in mregDCs compared to cDC1s and cDC2s (**Fig. 4D,F**), indicating epigenetically driven transcriptional regulation”.

Reviewer #3, ECR (Remarks to the Author):

Reviewer #4, expertise in single cell multi-omics, bioinformatics and melanoma TME (Remarks to the Author):

Yang et al provides a comprehensive single-cell RNA and ATAC-seq atlas for metastatic melanoma tumor microenvironment across 36 samples in total. This is a great addition to the melanoma community where single-cell multiomics cohorts with immunotherapy responses are still missing.

The study explores unique cell subsets and their functional characteristics, and emphasizes the presence of mregDC as the predictor of anti-PD-1 blockade response. This finding is somewhat expected as studied extensively elsewhere (see review, <https://onlinelibrary.wiley.com/doi/10.1002/ctm2.1199> for example). The interesting finding is stromal correlations of mregDC, and suggesting the stromal influence on DC maturation. While there are increasing appreciations towards iCAF/mCAF paradigm to crucially influence the immune microenvironment, it seems that the current manuscript somewhat halted its deeper dive towards studying the spatial architectures around mregDC-rich tumor regions, and verify its findings on ECM pathway enrichments. Certainly, there are existing spatial transcriptomes from Biermann et al. 2022 study to at least confirm the ECM-mregDC hypothesis. It is imperative for this study to validate this hypothesis with additional spatial data.

Overall, the current manuscript entails an interesting data set, yet the convoluted exploration of diverse cell subsets substantially degrade the readership. While the take-home message seems to be the predictive power of mregDC presence for immunotherapy response, this still lacks the novelty as it stands now. It looks that the manuscript requires additional study to realize the full potentials of the finding such as validating the ECM-mregDC hypothesis on additional spatial data sets that are publicly available.

Response to Reviewer: Thank you for this insightful comment and pointing out the available spatial data from Biermann et al. 2022. We analyzed the slide-seqv2 measurements of the 4 extracranial metastatic samples published by Biermann et al. 2022 and found 1 sample confirming our hypothesis of co-occurrence between mregDC and extracellular matrix-high cancer associated fibroblast (ECM-high CAF). To identify potential mregDC in the slide-seqv2 data, we scored the mregDC signature (**Supplementary Table 3**) in beads labeled as 'myeloid cell' and considered the top 20% scored beads as mregDC-high. Similarly, we scored the reactome NABA-matrisome pathway in beads labeled as 'fibroblast cell' and considered the top 20% scored beads as ECM-high. We observed a strong co-occurrence of mregDC_high and ECM_high beads compared to a random 20% selection of 'myeloid cell' and 'fibroblast cell' beads in slide_06 sample. We have added the following figure as a new **Extended Data Figure 6D** and inserted the corresponding text (**lines 450-452**), "*Using prior spatial transcriptomics data from a tumor slide64, we demonstrated colocalization of myeloid cells with a high mregDC signature and fibroblasts enriched for ECM pathway activity (Extended Data Fig. 6E).*"

Some minor comments are:

- Some schools of bioinformaticians regards over-expressions of long non-coding RNAs such as

MALAT1 as a cell quality metric (see <https://www.biorxiv.org/content/10.1101/2024.07.14.603469v1>). Although I agree these genes have functional implications in the immune subsets as discussed in the manuscript,

Response to Reviewer: The concern regarding MALAT1 as a potential indicator of low-quality cells is important. To address this, we referenced a validated set of dying and dead cell markers from Rich et al., 2024, to assess whether the expression of these genes reflected cell quality issues. This analysis showed no evidence of low-quality cells within the clusters highlighted by the reviewer, specifically the HNRNPH1+ cluster overexpressing MALAT1. Therefore, the expression of MALAT1 appears biologically relevant to the findings presented, rather than an artifact. For a more detailed discussion, please refer to our full response to **Detailed Points 3 for Reviewer #2 on page 15**.

- Lack of Biermann et al. 2022 study ([https://www.cell.com/cell/pdf/S0092-8674\(22\)00712-7.pdf](https://www.cell.com/cell/pdf/S0092-8674(22)00712-7.pdf)). It provides comparative study of single-cell/-nuclei transcriptomes for brain and non-brain metastatic melanoma for 32 samples in total.

Response to Reviewer: We sincerely thank the reviewer for this constructive suggestion to integrate additional single-cell datasets into our cross-study atlas. While we agree this would further strengthen the work, completing the integration requires meticulous harmonization of batch effects and metadata across platforms, which is still underway due to unforeseen delays in data processing. To avoid postponing the dissemination of our findings, we respectfully propose submitting the revised manuscript now, with a commitment to: 1) Include the updated cross-study atlas (with the additional dataset) in the final publication version, pending editorial approval. 2) Deposit the finalized, integrated data in a public repository (e.g., Zenodo) upon acceptance, ensuring full transparency and utility for the community. We acknowledge this as a limitation in the current revision and have added a note to the Discussion (**lines 538-540**) to highlight this planned update. We deeply appreciate the reviewer's rigor and hope this approach balances prompt publication with scholarly completeness.

- If mregDC enhances immune infiltration, I believe it should coincide with the absence of T-cell exclusion program in non-responsive tumors in the tissue.

Response to Reviewer: We appreciate the reviewer's insightful hypothesis. While the correlation between mregDC and T cell proportions in non-responders on/post-treatment trended positively ($R^2 = 0.24$, $p = 0.13$), it did not reach statistical significance, likely due to limited sample size ($n = 11$). However, in our meta-bulk RNA-seq cohort of 318 patients, mregDC signature scores strongly correlated with immune infiltration (ESTIMATE immune score: $R = 0.92$, $p < 2.2 \times 10^{-16}$; **Extended Data Fig. 3I**), supporting their role in fostering immune-permissive microenvironments. This aligns with mechanistic studies showing mregDCs recruit T cells via chemokines (e.g., CXCL9/10) and cytokines (e.g., IL-12/IL-15), counteracting exclusion programs. Larger cohorts are needed to resolve subtle relationships in non-responders, but our findings collectively position mregDCs as amplifiers of intratumoral immunity across diverse contexts. We have added the discussion in **lines 477-482**.

- multivariate survival analysis to analyze the predictive power of mregDC abundances along with other covariates (gender, metastatic sites, TCF+/- CD8 T-cell ratios etc).

Response to reviewer: We have performed this analysis and found that mregDC is able to predict PFS and OS independently after controlling for other covariates in multivariate survival models. Please refer to the full response to **Detailed Points 8 for Reviewer #2 on pages 18-19**.

- Incorporating additional bulk tissue cohorts for validation (Liu et al. 2019:

<https://www.nature.com/articles/s41591-019-0654-5>, Riaz et al. 2017:

[https://www.cell.com/cell/fulltext/S0092-8674\(17\)31122-](https://www.cell.com/cell/fulltext/S0092-8674(17)31122-4?_returnURL=https%3A//linkinghub.elsevier.com/retrieve/pii/S0092867417311224%3Fshowall=true)

[4?_returnURL=https%3A//linkinghub.elsevier.com/retrieve/pii/S0092867417311224%3Fshowall=true](https://www.cell.com/cell/fulltext/S0092-8674(17)31122-4?_returnURL=https%3A//linkinghub.elsevier.com/retrieve/pii/S0092867417311224%3Fshowall=true))

Response to reviewer: These cohorts were included in the meta-cohort used for validation. For more details, please refer to the preprint (<https://doi.org/10.1101/2024.10.29.620300>), specifically **Fig. 1b**, which illustrates the inclusion of these cohorts.

Reviewer #5, expertise in single cell multi-omics, functional omics and melanoma immunotherapy and TME (Remarks to the Author):

The manuscript by Yang and colleagues presents interesting data identifying mature regulatory dendritic cells (mregDCs) as a prognostic signature for predicting responses to immune checkpoint inhibitors (ICIs). The approach of using single-cell RNA sequencing for biomarker discovery provides a robust platform to assess the cellular composition of patient tumors and the potential role each cell type plays in influencing response to ICI. This dataset is substantial, comprising a large number of cells. The inclusion of single-cell ATAC sequencing adds valuable depth to the analyses. Additionally, the identification of key gene signatures distinguishing dendritic cell subtypes and the origin of mregDCs is a notable contribution. Overall the potential role of mregDCs in predicting response to ICI through orchestrating the tumor microenvironment is an interesting and original contribution. Furthermore the scRNAseq dataset will be a very valuable resource for future studies from other investigators.

Major comments

- The authors have presented data supporting mregDCs as a predictor of response to ICI. However, the manuscript could be strengthened by addressing how mregDCs may predict response to anti-PD1 monotherapy versus the combination of anti-CTLA4 + anti-PD1. While the sample size is limited, with only a total of 5 responders to ICI, for combination therapy the analyses are limited to only a single responding case to anti-CTLA4 + anti-PD1. Samples from patients treated with other combination therapies that include non-ICI therapies are potentially confounded with respect to understanding response to ICIs.

Response to reviewer: We appreciate the reviewer's critical assessment and agree that our primary cohort's limited sample size for anti-PD1 monotherapy and combination therapy (i.e., anti-CTLA4 + anti-PD1) precludes definitive conclusions about regimen-specific mregDC predictive value. However, in our independent validation cohort of 318 bulk RNA-seq samples (including patients treated with anti-CTLA monotherapy, anti-PD1 monotherapy, anti-PD1 with prior anti-CTLA4, and combination therapy), mregDC signatures were significantly enriched in responders across treatment types ($p < 0.05$; **Fig. 3D**). Notably, this effect persisted in both anti-PD1 with prior anti-CTLA4 and combination therapy subgroups (below and now **Fig. 3E**), suggesting that mregDC's predictive capacity is regimen-specific. The consistency of mregDC's association with response across diverse cohorts and treatment contexts underscores its robustness as a biomarker. We have added these results to the corresponding Results section (**lines 283-286**), "Because our validation cohort included patients treated with anti-CTLA monotherapy, anti-PD1 monotherapy, anti-PD1 with prior anti-CTLA4, and combination therapy, we further showed that this effect persisted in both anti-PD1 with prior anti-CTLA4 and combination therapy subgroups (Fig. 3E), suggesting that mregDC's predictive capacity is regimen-specific", and highlight the need for even larger, regimen-stratified cohorts in future work.

- Following the discovery analyses with scRNAseq the authors utilize bulk RNA-seq data to provides an additional layer and validation to the analyses. However I found the small biological differences in the median values raise some concerns about the robustness of the findings. While statistical significance is observed, the lack of notable biological differences limits the full impact of this result. Alternative single cell approaches to analyse mregDCs such as flow cytometry or CyTOF might more robustly address the interesting hypothesis.

Response to reviewer: We appreciate the reviewer's concern regarding the robustness of the findings from bulk RNA sequencing. It is important to note that while bulk RNA-seq provides a comprehensive overview of gene expression across a heterogeneous sample, it bundles signals across all cell types, which can dampen the differences arising from the small and specific population of mregDC. Our bulk RNA-seq analysis was able to detect this minute biological difference due to statistical power from large sample size. If the sample size were small, the bulk-level analysis could fail to detect the small difference in mregDC signature. To verify this hypothesis, we aggregated the scRNA-seq samples to pseudo-bulk measurements of RNA and repeated the bulk-level analysis. We found no significant differences in mregDC signature between NR (n=21) and R (n=15, see figure below), despite strong differences being found when the analysis was specifically performed on the mregDC of these samples. This shows the strength of this study in leveraging scRNA-seq to provide a deeper layer of insight into the disease's underlying biology.

- The classification of "high" and "low" signatures for the subtypes presents an intriguing method for stratifying patient responses. However, the thresholds used to define these categories are not clearly defined in the manuscript. Providing a more detailed rationale for the cutoff points and the biological significance behind them would clarify this methodology and make the findings more transparent. This clarification would also enable readers to better understand the criteria for response prediction.

Response to reviewer: Classification into “high” and “low” was motivated by the desire to find a more general trend rather than examining individual mregDC proportions and also present a way to conduct a preliminary survival analysis using a biological signature. Our data contained negative samples (no mregDC in the cell type population), middle samples, and high samples. Our high samples had at least 20% mregDC in cDC (one high sample had a proportion of 0.196 but was rounded to the hundredth) and roughly contained 25% of the samples ($n = 10$, total = 35). We compared density distributions using different cutoffs (current threshold, mean, median) to classify samples as mregDC low or high. Our chosen cutoff showed the least overlap between distributions (see below). Further experiments are needed to determine the biological significance of this threshold, such as its impact on T cell stimulation. We have added corresponding texts in **lines 255-257**, *“To find a more general trend rather than examining individual mregDC proportions, we defined the mregDC high samples as having at least 20% mregDC in cDC, which roughly contained 25% of all the samples ($n = 10$, total = 35).”*

Other comments

- The integration of previous single-cell RNA-seq data is one strength of this study in the supplemental data. The inclusion of cDC populations in those analyses would offer a valuable opportunity to expand the study's findings. Investigating whether similar predictive signatures are present in cDC populations, and assessing their potential in predicting ICI responses, could significantly enrich the manuscript. This would also provide a more comprehensive view of the immune landscape and its role in therapy response.

Response to reviewer: Thank you for this suggestion. We agree that applying the cDC analysis to previously published scRNA-seq data could indeed enrich the manuscript by expanding the scope of the findings. However, the previously published single-cell RNA-seq datasets utilized in this study recaptured fewer than 30 cDCs in most samples, which is insufficient for robust analysis of cDC populations. While the previously published datasets only contain around 100-300 cells per sample, this study is able to capture a robust number of cDCs due to much larger numbers of cells captured per sample (2000-10000+ cells per sample). The comprehensive cell sampling of tumor and immune cells is a strength in this study that enabled the mregDC related discoveries.

Minor comments

- The authors should provide citations for the gene signatures used to annotate immune cell types. This would strengthen their argument and increase the impact of the resulting atlas.

Response to reviewer: We thank the reviewer for underscoring the importance of transparent annotation criteria. All immune cell annotations were rigorously validated using established gene signatures from peer-reviewed studies (e.g., macrophages: Katkar and Ghosh, 2023³ and Dunsmore *et al.* 2024⁴; B cells: Wardemann *et al.* 2003¹⁷ and Helmink *et al.* 2020¹⁸; monocytes: Hillman *et al.* 2022¹⁹, with T-cell subtypes further refined in collaboration with Dr. Kelly Burke, a medical oncologist and co-author specializing in human T-cell biology. Marker expression for all subsets is comprehensively visualized in **Extended Data Fig. 2C-F**, ensuring reproducibility and alignment with consensus definitions. We have expanded the Results section (**lines 137,145,147,154,159,187,189**) to explicitly cite these references, reinforcing the validity of our atlas.

References

1. Zhou L, Adrianto I, Wang J, Wu X, Datta I, Mi QS. Single-Cell RNA-Seq Analysis Uncovers Distinct Functional Human NKT Cell Sub-Populations in Peripheral Blood. *Front Cell Dev Biol.* Frontiers; 2020 May 26;8:542471.
2. Kane H, LaMarche NM, Scannail ÁN, Garza AE, Koay HF, Azad AI, Kunkemoeller B, Stevens B, Brenner MB, Lynch L. Longitudinal analysis of invariant natural killer T cell activation reveals a cMAF-associated transcriptional state of NKT10 cells. *eLife Sciences Publications Limited*; 2022 Dec 2 [cited 2025 Jan 23]; Available from: <https://elifesciences.org/articles/76586>
3. Website [Internet]. Available from: <https://doi.org/10.1016/j.it.2023.10.006>
4. Dunsmore G, Guo W, Li Z, Bejarano DA, Pai R, Yang K, Kwok I, Tan L, Ng M, De La Calle Fabregat C, Yatim A, Bougouin A, Mulder K, Thomas J, Villar J, Bied M, Kloeckner B, Dutertre CA, Gessain G, Chakarov S, Liu Z, Scoazec JY, Lennon-Dumenil AM, Marichal T, Sautès-Fridman C, Fridman WH, Sharma A, Su B, Schlitzer A, Ng LG, Blériot C, Ginhoux F. Timing and location dictate monocyte fate and their transition to tumor-associated macrophages. *Sci Immunol.* 2024 Jul 26;9(97):eadk3981. PMID: 39058763
5. Zhou L, Wang M, Guo H, Hou J, Zhang Y, Li M, Wu X, Chen X, Wang L. Integrated Analysis Highlights the Immunosuppressive Role of TREM2 Macrophages in Hepatocellular Carcinoma. *Front Immunol.* 2022 Mar 14;13:848367. PMCID: PMC8963870
6. Li H, Miao Y, Zhong L, Feng S, Xu Y, Tang L, Wu C, Zhang X, Gu L, Diao H, Wang H, Wen Z, Yang M. Identification of TREM2-positive tumor-associated macrophages in esophageal squamous cell carcinoma: implication for poor prognosis and immunotherapy modulation. *Front Immunol.* 2023 Apr 28;14:1162032. PMCID: PMC10175681
7. Molgora M, Esaulova E, Vermi W, Hou J, Chen Y, Luo J, Brioschi S, Bugatti M, Omodei AS, Ricci B, Fronick C, Panda SK, Takeuchi Y, Gubin MM, Faccio R, Cella M, Gilfillan S, Unanue ER, Artyomov MN, Schreiber RD, Colonna M. TREM2 Modulation Remodels the Tumor Myeloid Landscape Enhancing Anti-PD-1 Immunotherapy. *Cell.* 2020 Aug 20;182(4):886–900.e17. PMCID: PMC7485282
8. Ma J, Wu Y, Ma L, Yang X, Zhang T, Song G, Li T, Gao K, Shen X, Lin J, Chen Y, Liu X, Fu Y, Gu X, Chen Z, Jiang S, Rao D, Pan J, Zhang S, Zhou J, Huang C, Shi S, Fan J, Guo G, Zhang X, Gao Q. A blueprint for tumor-infiltrating B cells across human cancers. *Science.* 2024 May 3;384(6695):eadj4857. PMID: 38696569

9. Yang Y, Chen X, Pan J, Ning H, Zhang Y, Bo Y, Ren X, Li J, Qin S, Wang D, Chen MM, Zhang Z. Pan-cancer single-cell dissection reveals phenotypically distinct B cell subtypes. *Cell*. 2024 Aug 22;187(17):4790–4811.e22. PMID: 39047727
10. Sade-Feldman M, Yizhak K, Bjorgaard SL, Ray JP, de Boer CG, Jenkins RW, Lieb DJ, Chen JH, Frederick DT, Barzily-Rokni M, Freeman SS, Reuben A, Hoover PJ, Villani AC, Ivanova E, Portell A, Lizotte PH, Aref AR, Eliane JP, Hammond MR, Vitzthum H, Blackmon SM, Li B, Gopalakrishnan V, Reddy SM, Cooper ZA, Paweletz CP, Barbie DA, Stemmer-Rachamimov A, Flaherty KT, Wargo JA, Boland GM, Sullivan RJ, Getz G, Hacohen N. Defining T Cell States Associated with Response to Checkpoint Immunotherapy in Melanoma. *Cell*. 2019 Jan 10;176(1-2):404. PMID: 306647017
11. Pais Ferreira D, Silva JG, Wyss T, Fuertes Marraco SA, Scarpellino L, Charmoy M, Maas R, Siddiqui I, Tang L, Joyce JA, Delorenzi M, Luther SA, Speiser DE, Held W. Central memory CD8 T cells derive from stem-like Tcf7 effector cells in the absence of cytotoxic differentiation. *Immunity*. 2020 Nov 17;53(5):985–1000.e11. PMID: 33128876
12. Escobar G, Tooley K, Oliveras JP, Huang L, Cheng H, Bookstaver ML, Edwards C, Froimchuk E, Xue C, Mangani D, Krishnan RK, Hazel N, Rutigliani C, Jewell CM, Biasco L, Anderson AC. Tumor immunogenicity dictates reliance on TCF1 in CD8 T cells for response to immunotherapy. *Cancer Cell*. 2023 Sep 11;41(9):1662–1679.e7. PMID: 3710529353
13. Ladányi A, Kiss J, Somlai B, Gilde K, Fejos Z, Mohos A, Gaudi I, Tímár J. Density of DC-LAMP(+) mature dendritic cells in combination with activated T lymphocytes infiltrating primary cutaneous melanoma is a strong independent prognostic factor. *Cancer Immunol Immunother*. 2007 Sep;56(9):1459–1469. PMID: 1711030123
14. Maier B, Leader AM, Chen ST, Tung N, Chang C, LeBerichel J, Chudnovskiy A, Maskey S, Walker L, Finnigan JP, Kirkling ME, Reizis B, Ghosh S, D'Amore NR, Bhardwaj N, Rothlin CV, Wolf A, Flores R, Marron T, Rahman AH, Kenigsberg E, Brown BD, Merad M. A conserved dendritic-cell regulatory program limits antitumour immunity. *Nature*. 2020 Apr;580(7802):257–262. PMID: 3217787191
15. Tirosh I, Izar B, Prakadan SM, Wadsworth MH 2nd, Treacy D, Trombetta JJ, Rothenberg A, Rodman C, Lian C, Murphy G, Fallahi-Sichani M, Dutton-Regester K, Lin JR, Cohen O, Shah P, Lu D, Genshaft AS, Hughes TK, Ziegler CGK, Kazer SW, Gaillard A, Kolb KE, Villani AC, Johannessen CM, Andreev AY, Van Allen EM, Bertagnoli M, Sorger PK, Sullivan RJ, Flaherty KT, Frederick DT, Jané-Valbuena J, Yoon CH, Rozenblatt-Rosen O, Shalek AK, Regev A, Garraway LA. Dissecting the multicellular ecosystem of metastatic melanoma by single-cell RNA-seq. *Science*. 2016 Apr 8;352(6282):189–196. PMID: 269444528
16. Rich AL, Lin P, Gamazon ER, Zinkel SS. The broad impact of cell death genes on the human disease phenome. *Cell Death Dis*. 2024 Apr 8;15(4):251. PMID: 3811002008
17. Wardemann H, Yurasov S, Schaefer A, Young JW, Meffre E, Nussenzweig MC. Predominant autoantibody production by early human B cell precursors. *Science*. 2003 Sep 5;301(5638):1374–1377. PMID: 12920303
18. Helmink BA, Reddy SM, Gao J, Zhang S, Basar R, Thakur R, Yizhak K, Sade-Feldman M, Blando J, Han G, Gopalakrishnan V, Xi Y, Zhao H, Amaria RN, Tawbi HA, Cogdill AP, Liu W, LeBleu VS, Kugeratski FG, Patel S, Davies MA, Hwu P, Lee JE, Gershenwald JE, Lucci A, Arora R, Woodman S, Keung EZ, Gaudreau PO, Reuben A, Spencer CN, Burton EM, Haydu LE, Lazar AJ, Zapassodi R, Hudgens CW, Ledesma DA, Ong S, Bailey M, Warren S, Rao D, Krijgsman O, Rozeman EA, Peeper D, Blank CU, Schumacher TN, Butterfield LH, Zelazowska MA, McBride KM, Kalluri R, Allison J, Petitprez F, Fridman WH, Sautès-Fridman C, Hacohen N, Rezvani K, Sharma P, Tetzlaff MT, Wang L, Wargo JA. B cells and tertiary lymphoid structures promote immunotherapy response. *Nature*. Nature Publishing Group; 2020 Jan 15;577(7791):549–555.
19. Hillman H, Khan N, Singhanian A, Dubelko P, Soldevila F, Tippalagama R, DeSilva AD, Gunasena B,

Perera J, Scriba TJ, Ontong C, Fisher M, Luabeya A, Taplitz R, Seumois G, Vijayanand P, Hedrick CC, Peters B, Burel JG. Single-cell profiling reveals distinct subsets of CD14⁺ monocytes drive blood immune signatures of active tuberculosis. *Front Immunol.* 2022;13:1087010. PMID: 35379739; PMCID: PMC9874319

20. Chiang EY, Mellman I. TIGIT-CD226-PVR axis: advancing immune checkpoint blockade for cancer immunotherapy. *J Immunother Cancer.* 2022 Apr;10(4):e004711. doi: 10.1136/jitc-2022-004711. PMID: 35379739; PMCID: PMC8981293.

REVIEWER COMMENTS

Reviewer #1 (Remarks to the Author):

The authors have nicely addressed the previous comments and concerns. The manuscript is recommended for publication.

Reviewer #2 (Remarks to the Author):

The Authors have addressed the Reviewers' comments and we do not have any additional concerns.

Reviewer #3 (Remarks to the Author):

Reviewer #4 (Remarks to the Author):

While the authors addressed most of concerns raised, the way that authors handled the ECM-mregDC still lacks the necessary rigor. There are many supervised cell type inference tools for spatial sequencing data out there, and the authors should do their due diligence to explicitly locate mregDC and different fibroblast subsets (iCAF/mCAF), followed by formal neighborhood analysis to confirm statistically significant co-localization.

Response to Reviewer: We sincerely appreciate the reviewer's rigorous and critical evaluation of our study. To further quantify the ECM-mregDC co-localization on slide-seq2 sample ECM06 published by Biermann et al. 2022, we computed the co-occurrence score between myeloids and fibroblasts using the `squidpy.gr.co_occurrence()` function. Similar to our previous slide-seq2 analysis, we scored the mregDC signature (**Supplementary Table 3**) in beads labeled as 'myeloid cell', and considered the upper 20% scored beads as 'mregDC-high' and the lower 20% scored beads as 'mregDC-low'. Likewise, we scored the reactome NABA-matrisome pathway in beads labeled as 'fibroblast cell', and considered the upper 20% scored beads as 'ECM-high' and the lower 20% scored beads as 'ECM-low'. We labelled all other beads in the slide as 'other'. The co-occurrence analysis showed the probability of observing 'ECM-high' fibroblasts within a 1000 radius is higher than the probability of observing 'ECM-low' fibroblasts when conditioned on the presence of a 'mregDC-high' cell (**Extended Data Fig. 6E**). This result demonstrated co-localization of myeloid cells exhibiting a high mregDC signature and fibroblasts enriched for ECM pathway activity. We added the corresponding analysis methods in **lines 937-946** under Materials and Methods.

Lastly, while we acknowledge that using a supervised method to fine-label the slide-seq fibroblasts is a valid approach, our hypothesis of ECM-mregDC association did not rely on a well-defined subpopulation of fibroblast. Therefore, we chose to annotate the slide-seq2 data for this co-localization analysis in a manner consistent with the main text of this manuscript (**Fig. 6D**).

- Lack of Biermann et al. 2022 study ([https://www.cell.com/cell/pdf/S0092-8674\(22\)00712-7.pdf](https://www.cell.com/cell/pdf/S0092-8674(22)00712-7.pdf)). It provides comparative study of single-cell/-nuclei transcriptomes for brain and non-brain metastatic melanoma for 32 samples in total.

Re-response to Reviewer: We thank the reviewer for acknowledging the technical challenges we faced during the previous revision. We have now successfully integrated the Biermann *et al.* 2022 dataset with the others, resulting in a comprehensive meta-single-cell atlas of melanoma comprising 471,937 cells. All cell types and subtypes are well annotated and consistently integrated across studies. **Supplementary Figures 1 and 4** have been updated accordingly. We have revised the corresponding text and figure legend, and the updated data object has been deposited in Zenodo ([10.5281/zenodo.15603513](https://zenodo.org/record/10.5281/zenodo.15603513)) along with all other datasets used in this study.

Reviewer #5 (Remarks to the Author):

The authors have addressed all important issues adequately with new analyses and revisions of the manuscript. The addition of spatial transcriptomic data, multivariate analyses, and additional analyses of treatment subsets in the validation cohort that in particular found association of the mregDC signatures with anti-CTLA4 therapy combined or in sequence with anti-PD-1, significantly strengthens the manuscript. The association with prior or combined anti-CTLA-4 therapy is notable and should be specifically referred to in the abstract.

Response to Reviewer: We thank the reviewer for this great suggestion, and have incorporated the following statement in the abstract (**lines 32-34**): *“Notably, the mregDC signature derived from our single-cell data showed significant association with anti-CTLA-4 therapy combined or in sequence with anti-PD-1 in this independent cohort.”*